Embracing heterogeneity: coalescing the Tree of Life and the future of phylogenomics

http://orcid.org/0000-0001-5889-2767 Bravo Gustavo A. 1 gustavo_bravo@fas.harvard.edu
http://orcid.org/0000-0003-1842-9297 Antonelli Alexandre 1 2 3 4
http://orcid.org/0000-0003-2341-2705 Bacon Christine D. 2 3
http://orcid.org/0000-0002-5816-4345 Bartoszek Krzysztof 5
http://orcid.org/0000-0002-6304-9827 Blom Mozes P. K. 6
Huynh Stella 7
http://orcid.org/0000-0002-9686-5871 Jones Graham 3
http://orcid.org/0000-0002-6567-4853 Knowles L. Lacey 8
Lamichhaney Sangeet 1
http://orcid.org/0000-0002-1960-9508 Marcussen Thomas 9
Morlon Hélène 10
http://orcid.org/0000-0003-3288-6769 Nakhleh Luay K. 11
Oxelman Bengt 2 3
Pfeil Bernard 3
http://orcid.org/0000-0002-3555-3188 Schliep Alexander 12
http://orcid.org/0000-0002-1259-3363 Wahlberg Niklas 13
http://orcid.org/0000-0002-8779-2607 Werneck Fernanda P. 14
http://orcid.org/0000-0002-6935-1517 Wiedenhoeft John 12 15
Willows-Munro Sandi 16
http://orcid.org/0000-0003-2535-6217 Edwards Scott V. 1 17
1 Department of Organismic and Evolutionary Biology, Museum of Comparative Zoology, Harvard University , Cambridge, MA , USA
2 Gothenburg Global Biodiversity Centre , Göteborg , Sweden
3 Department of Biological and Environmental Sciences, University of Gothenburg , Göteborg , Sweden
4 Gothenburg Botanical Garden , Göteborg , Sweden
5 Department of Computer and Information Science, Linköping University , Linköping , Sweden
6 Department of Bioinformatics and Genetics, Swedish Museum of Natural History , Stockholm , Sweden
7 Institut de Biologie, Université de Neuchâtel , Neuchâtel , Switzerland
8 Department of Ecology and Evolutionary Biology, University of Michigan , Ann Arbor, MI , USA
9 Centre for Ecological and Evolutionary Synthesis, University of Oslo , Oslo , Norway
10 Institut de Biologie, Ecole Normale Supérieure de Paris , Paris , France
11 Department of Computer Science, Rice University , Houston, TX , USA
12 Department of Computer Science and Engineering, Chalmers University of Technology and University of Gothenburg , Göteborg , Sweden
13 Department of Biology, Lund University , Lund , Sweden
14 Coordenação de Biodiversidade, Programa de Coleções Científicas Biológicas, Instituto Nacional de Pesquisa da Amazônia , Manaus, AM , Brazil
15 Department of Computer Science, Rutgers University , Piscataway, NJ , USA
16 School of Life Sciences, University of Kwazulu-Natal , Pietermaritzburg , South Africa
17 Gothenburg Centre for Advanced Studies in Science and Technology, Chalmers University of Technology and University of Gothenburg , Göteborg , Sweden
Creevey Chris
Electronic publication date: 2019 Feb 14
Publication date: 2019
Volume: 7
Electronic Location ID: e6399
Received 2018 Feb 5; Accepted 2019 Jan 7
Copyright: © 2019 Bravo et al.
Copyright year: 2019
Copyright holder: Bravo et al.
License: This is an open access article distributed under the terms of the Creative Commons Attribution License, which permits unrestricted use, distribution, reproduction and adaptation in any medium and for any purpose provided that it is properly attributed. For attribution, the original author(s), title, publication source (PeerJ) and either DOI or URL of the article must be cited.
License URL: https://creativecommons.org/licenses/by/4.0/

Keywords: Gene flow, Genome, Multispecies coalescent model, Retroelement, Speciation, Transcriptome

Funding: Chalmers University of Technology University of Gothenburg Swedish Research Council (Bengt Oxelman, Alexandre Antonelli) U.S. National Science Foundation European Research Council under the European Union’s Seventh Framework Programme FP/2007-2013, ERC Grant Agreement n. 331024 Swedish Foundation for Strategic Research Wallenberg Academy Fellowship Conselho Nacional de Desenvolvimento Científico e Tecnológico—CNPq Partnerships for Enhanced Engagement in Research from the U.S. National Academy of Sciences U.S. Agency of International Development—PEER NAS/USAID L’Oreal-Unesco For Women in Science Program The Gothenburg Center for Advanced Studies (GoCas) workshop ‘Origins of Biodiversity’ was funded by Chalmers University of Technology and the University of Gothenburg. The authors are supported by scholarships or research grants from the following agencies: Swedish Research Council (Bengt Oxelman, Alexandre Antonelli); U.S. National Science Foundation; European Research Council under the European Union’s Seventh Framework Programme (FP/2007-2013, ERC Grant Agreement n. 331024 to Alexandre Antonelli); Swedish Foundation for Strategic Research; Wallenberg Academy Fellowship (Alexandre Antonelli); Conselho Nacional de Desenvolvimento Científico e Tecnológico—CNPq (Fernanda P. Werneck); Partnerships for Enhanced Engagement in Research from the U.S. National Academy of Sciences (Fernanda P. Werneck); U.S. Agency of International Development—PEER NAS/USAID (Fernanda P. Werneck); and L’Oreal-Unesco For Women in Science Program (Fernanda P. Werneck). The funders had no role in study design, data collection and analysis, decision to publish, or preparation of the manuscript.

==============================
Building the Tree of Life (ToL) is a major challenge of modern biology, requiring advances in cyberinfrastructure, data collection, theory, and more. Here, we argue that phylogenomics stands to benefit by embracing the many heterogeneous genomic signals emerging from the first decade of large-scale phylogenetic analysis spawned by high-throughput sequencing (HTS). Such signals include those most commonly encountered in phylogenomic datasets, such as incomplete lineage sorting, but also those reticulate processes emerging with greater frequency, such as recombination and introgression. Here we focus specifically on how phylogenetic methods can accommodate the heterogeneity incurred by such population genetic processes; we do not discuss phylogenetic methods that ignore such processes, such as concatenation or supermatrix approaches or supertrees. We suggest that methods of data acquisition and the types of markers used in phylogenomics will remain restricted until a posteriori methods of marker choice are made possible with routine whole-genome sequencing of taxa of interest. We discuss limitations and potential extensions of a model supporting innovation in phylogenomics today, the multispecies coalescent model (MSC). Macroevolutionary models that use phylogenies, such as character mapping, often ignore the heterogeneity on which building phylogenies increasingly rely and suggest that assimilating such heterogeneity is an important goal moving forward. Finally, we argue that an integrative cyberinfrastructure linking all steps of the process of building the ToL, from specimen acquisition in the field to publication and tracking of phylogenomic data, as well as a culture that values contributors at each step, are essential for progress.

Introduction

Phylogenomics has been greatly enriched with the introduction of high-throughput sequencing (HTS) and increased breadth of phylogenomic sampling, which have allowed researchers interested in the Tree of Life (ToL) to scale up in several dimensions and placing both fields squarely in the era of ‘big data’. Additionally, conceptual advances and improvements of statistical models used to analyze these data are helping bridge what some have perceived as a gap between phylogenetics and phylogeography (e.g., Felsenstein, 1988; Huson & Bryant, 2006; Edwards et al., 2016a). However, as datasets become larger, researchers are inevitably faced with a plethora of heterogeneous signals across different genomic regions that often depart from a dichotomously-branching phylogeny (Kunin, Goldovsky & Darzentas, 2005; Jeffroy et al., 2006; Mallet, Besansky & Hahn, 2015). These signals cover an increasingly large array of biological processes at the level of genes and genomes, as well as individual organisms and populations, including processes such as recombination, hybridization, gene flow, and polyploidization; they can be thought of as conflicting, but in truth, they are simply a record of the singular history that we commonly refer to as the ToL (King & Rokas, 2017). One of the grand challenges of evolutionary biology is deciphering this history, whether at the level of genes, populations, species, or genomes, however, currently phylogenomicists have not yet determined how to fully exploit the diverse signals in molecular data.

In this perspective piece, our goal is to highlight opportunities for the field of phylogenomics to overcome conceptual and practical challenges as we navigate our way through the era of big data. We argue that conceptually and analytically embracing heterogeneity generated by population-level processes and associated with different histories across the genome will lead to increased insight into the ToL and its underlying processes. Because major theoretical advances in phylogenomics and population genetics resulting from the advent of genome-scale data have been reviewed elsewhere (e.g., Delsuc, Brinkmann & Philippe, 2005; Degnan & Rosenberg, 2009; Ellegren & Galtier, 2016; Payseur & Rieseberg, 2016; Hahn, 2018), here we focus on advances and challenges to phylogenomics specifically brought about by population genetic processes, which inevitably leads us to focus on the major conceptual framework dealing with such processes, the multispecies coalescent model (MSC). Likewise, advances in other methods such as supertrees (reviewed by Warnow, 2018), supermatrices (de Queiroz & Gatesy, 2007; Philippe et al., 2017), and partitioned analyses (e.g., Lanfear et al., 2014; Kainer & Lanfear, 2015) are not central to the objectives of this perspective piece and further details on those topics can be found elsewhere.

A key concept introduced by the scaling up from phylogeography to phylogenomics is the continuum of processes and analytical methods—the so-called phylogeography-phylogenetics continuum (Edwards et al., 2016a). We argue here that bridging this continuum is critical for advancing phylogenetics. This bridging can be done by either developing phylogenomic approaches that acknowledge and explicitly account for phylogeographic processes, or by determining the regions of parameter space (e.g., branch lengths in tree, level of gene flow) if any, where such within-species processes are not relevant. For example, the choice of markers in a given phylogenomics project is currently guided more by convenience and cost than by evaluating the biological properties and phylogenomic signals in those data; but comparisons of signals across various types of markers (e.g., transcriptomes, noncoding regions) reveal that marker choice is a critical step toward shedding light on the history of populations and unraveling potential processes underlying such history (Rokas et al., 2003; Cutter, 2013; Jarvis et al., 2014; Reddy et al., 2017). On the analysis side, we are in desperate need of methods that can handle the increasingly large data sets being produced by empiricists, but at the same time there is a desire to include increasingly diverse sources of signal in estimates of divergence times, biogeographic history, and models of diversification (Delsuc, Brinkmann & Philippe, 2005; Jeffroy et al., 2006; Kumar et al., 2012). Finding the balance between breadth, depth, and computational feasibility in project design and statistical analysis is crucial for the field today.

Additionally, we discuss ways in which data archiving and integration can benefit access to phylogenomic data and the contributions of phylogenomics to society. Are the priorities that society places on the many disciplines feeding into scientific efforts toward the ToL—fieldwork, museum collections, databases—appropriate for this grand mission? Although we cannot possibly answer all of these questions within the scope of this perspective, we hope to at least spur discussion on the wide range of field, laboratory, conceptual, and societal issues that allow phylogenomics to move forward.

We first describe the types of genomic data that researchers can generate to perform phylogenomic analyses and how those are more or less suitable for phylogenomic and phylogeographic analyses. We then discuss key concepts around the MSC and highlight the need to expand this model beyond its current limitations. We then discuss how the interplay between phylogenomics and macroevolution might strengthen our understanding of diversity patterns and offer suggestions as to how the community can overcome limitations posed by current methods and models in both fields. Finally, we discuss desired practices that, as a community, phylogenomicists, museum scientists, field biologists, bioinformaticians, and other scientists can embrace toward the goal of assembling the ToL.

Survey Methodology

During the ‘Origins of Biodiversity Workshop’ organized during May 15–19, 2017 by Chalmers University of Technology and the University of Gothenburg, Sweden, under the auspices of the Gothenburg Centre for Advanced Studies (GoCAS), we gathered scholars and students from several countries and scientific backgrounds to discuss future perspectives in the fields of phylogenomics and phylogeography. We spent one week sharing our recent experiences in these fields and outlining the topics presented here and continued remotely to complete this review. Our goal is not to provide a complete overview of phylogenomic and phylogeographic research, but rather present a number of conceptual and practical aspects that we feel are essential to keep the momentum that these fields have gained during recent times.

Data Generation and Data Types in Phylogenomics

The need for large-scale phylogenomic data

One of the fundamental challenges in evolutionary biology is to estimate a ToL for all species. The potential impact of such large phylogenies is reflected in their publication in the highest impact journals, but also in their broad contribution, which extends beyond big data, to methodological innovations, and downstream understanding of macroevolutionary processes (e.g., coalescent methods of species tree inference; accounting for hybridization and unsampled species or localities in datasets; understanding community or genome evolution through large-scale phylogenetics). Hence, the phylogenomics community is now placing a high priority on very large-scale trees, whether in terms of number of taxa, number of genes, or both. The current need for large phylogenies and the high priority placed on them by high-impact journals can also result in short-cuts, wherein extant species lacking any molecular data are often placed in trees based on current taxonomy (Jetz et al., 2012; Zanne et al., 2014; Faurby & Svenning, 2015), which can result in conflicts with more robust estimates based on actual data (Brown, Wang & Smith, 2017). At the same time, however, hypothesis-testing in areas such as macroevolution, macroecology, biodiversity, and systematics require these large-scale trees, even as they present challenges being built on high quality data. The phylogenetic knowledge on which we lay a foundation for downstream analyses must be robust, and therefore it is essential that the input phylogenetic hypotheses themselves are robust (Pyron, 2015). Indeed, the current bottlenecks in large-scale phylogenomic data do not appear to be the sequencing, but rather the compilation of high quality, well-curated genomic resources and the availability of adequate software and methods that can fuel phylogenomics for the next century (e.g., Global Genome Initiative, www.mnh.si.edu/ggi/).

Data quality

Genome-scale data in the form of multiple alignments and other homology statements are the foundation of phylogenomics. A major challenge is the difficulty of comprehensive quality checks of data, given that HTS datasets are so large. As researchers collect datasets consisting of thousands of alignments across scores of species, data quality is a serious concern that is left for detection and handling primarily by computer algorithms. In addition to inherent systematic errors in the data (Kocot et al., 2017), several examples of errors in phylogenomic data sets have been reported in the literature, including the use of unintended paralogous sequences in alignments (e.g., Struck, 2013); mistaking the genome sequence of one species for another (Philippe et al., 2011); and inclusion of genome sequence from parasites into the genome of the host (Kumar et al., 2013). However, the incidence of smaller errors in alignments that are not easily discerned from natural allelic variation, such as base miscalls or misplaced indels, are probably much more widespread than has been reported in the literature. Combined with the sensitivity of some phylogenomic datasets to individual loci or single nucleotide polymorphisms (SNPs) within loci (Shen, Hittinger & Rokas, 2017), such errors could have damaging consequences for phylogenomic studies, for the inference of topology and branch lengths of both gene trees and species phylogenies (Marcussen et al., 2014; Bleidorn, 2017). Furthermore, as phylogenomic datasets increase in size, it is likely that the accumulation of errors due to the use of different sequencing chemistries and sequencing depths (Quail et al., 2012; Goodwin, McPherson & McCombie, 2016) will ultimately influence phylogenetic inference. We predict that the impact of these errors will largely depend on the sampling breadth and taxonomic scale of each study, and whether the species phylogeny is a tree or a network.

Sequencing high-quality samples from well-archived voucher specimens is a good first step to increase reproducibility and alleviate issues related to sample identity (Peterson et al., 2007; Pleijel et al., 2008; Chakrabarty, 2010; Turney et al., 2015; Troudet et al., 2018). For individual phylogenomic studies, wholesale manual inspection of every locus is unsustainable (Irisarri et al., 2017), but spot checks of a subset of the data (e.g., 5–10% of the alignments) is a recommended best practice (Philippe et al., 2011) that is beginning to be encouraged in peer review and in published papers (Montague et al., 2014; Liu et al., 2017). Such checking is important not only for new data generated by a given study, but also for data downloaded from public repositories such as NCBI and OrthoMaM (Ranwez et al., 2007; Douzery et al., 2014), which are well known to contain errors (Wesche, Gaffney & Keightley, 2009). Because several databases do not include the raw sequence data it is often impossible to evaluate whether oddities may derive from poor sequencing. Robust pipelines for flagging poorly aligned sites or non-homologous sequences, based on existing tools or novel scripts such as Gblocks (Castresana, 2000; Talavera & Castresana, 2007) or TrimAl (Capella-Gutierrez, Silla-Martinez & Gabaldon, 2009) are gradually being put into practice (Marcussen et al., 2014; He et al., 2016; Irisarri et al., 2017).

Coding regions, whether derived from transcriptomes or whole-genome data, are particularly amenable to spot checking of alignments and to filtering out of low-quality data with bioinformatic pipelines (e.g., Dunn, Howinson & Zapata, 2013; Blom, 2015). Coding regions have the advantage of allowing amino acids to guide alignments, which is particularly useful for highly divergent sequences. Stop codons can help flag errors or genuine pseudogenes. Examining gene tree topologies is also widely used to detect paralogs in phylogenomic data (e.g., Betancur, Naylor & Ortí, 2014). Examining gene trees for aberrantly long branch lengths can also reveal misalignments (e.g., He et al., 2016); sensitivity analyses of methods for indirectly detecting errors in alignments are sorely needed.

Data generation and marker development

Genome reduction methods

A growing number of genome reduction methods are now providing empiricists with the means to generate genomic subsets suitable for phylogenetic and phylogeographic inference (reviewed by McCormack et al., 2013; Leaché & Oaks, 2017; Lemmon & Lemmon, 2013). For phylogenomics, most prominently featured are sequence-capture, focusing on highly conserved regions (e.g., Faircloth et al., 2012; Lemmon, Emme & Lemmon, 2012; reviewed by Jones & Good, 2016) and transcriptomes (e.g., Misof et al., 2014; Cohen et al., 2016; Fernández et al., 2014; Park et al., 2015; Simion et al., 2017; Irisarri et al., 2017), but phylogenomic trees have also been constructed based on restriction-digest methods that primarily focus on SNPs (Leaché, Chavez & Jones, 2015; Harvey et al., 2016) and analysis of transposable elements (e.g., Suh, Smeds & Ellegren, 2015). This diversity of marker types for phylogenetics should be celebrated, but each marker type brings with it a list of pros and cons. For example, many questions in the higher level phylogenetics of animals and plants have so far relied almost exclusively on transcriptome data. However, the uncritical use of transcriptomes in phylogenetics is not without caveats. At high taxonomic levels, coding regions can exhibit extreme levels of among-taxon base composition, sometimes resulting in strong violations of phylogenetic models (Romiguier et al., 2016; Romiguier & Roux, 2017). Coding regions can exhibit reduced levels of incomplete lineage sorting (ILS) compared to noncoding regions (Scally et al., 2012). Such reduced ILS could in fact be helpful in building complex phylogenies with rapid radiations (Edwards, 2009a), but it will certainly distort estimated branch lengths when coalescent methods, which assume neutrality, are used. In addition to yielding abundant sequence variations and SNPs, transcriptome data also yields information on the levels of expression of various genes, and in which tissue-specific genes are expressed. However, using these aspects of transcriptome data are less likely to be of use to phylogeneticists, precisely because specific genes are often tissue-specific and because expression data can exhibit high levels of variation among individuals, populations and species in space and time (Todd, Black & Gemmell, 2016). Such variation will certainly pose limitations for long term phylogenomic endeavors that will likely combine data collected originally for different purposes.

Although non-coding portions of the genome have been largely neglected in phylogenomics because they are difficult to align and analyze, we are now making progress in understanding their sequence evolution and how it might be informative for comparative purposes (Ulitsky, 2016; Edwards, Cloutier & Baker, 2017). For instance, studies on enhancer and promoter evolution in mammals have shown that despite low levels of sequence conservation, there is conservation of regulatory function and 3D structure across species that carries information for comparative purposes (Villar et al., 2015; Berthelot et al., 2018). The development of methods to infer and interpret the evolutionary history and phylogenetic signal of non-coding elements and 3D genome structure is a critical priority.

Single nucleotide polymorphisms have been advocated by some authors for higher level phylogenetics (Leaché & Oaks, 2017), but the available methods for analyzing such data are still extremely limited. For example, concatenation and two coalescent methods (SNAPP and SVD quartets: Bryant et al., 2012; Chifman & Kubatko, 2014) have recently been highlighted as the main methods available for phylogenomic analysis of SNPs (Leaché & Oaks, 2017). But each of these methods has its shortcomings. It is likely that concatenation of SNPs will be misleading for many of the same reasons that concatenating genes can be misleading, due to different gene histories (see section ‘Concepts and models in phylogenomics’ for further details; Kubatko & Degnan, 2007). SNAPP, a coalescent method suitable for analysis of SNPs (Bryant et al., 2012), works well only on relatively small data sets, and it is unclear how well SVD quartets performs on some data sets (Shi & Yang, 2018). Although SNPs do provide a helpful route around the often-violated assumption in coalescent models of no recombination within loci (Bryant et al., 2012), and are informative markers for phylogeography and population genetics (Brumfield et al., 2003), it remains to be seen how powerful they are at higher phylogenetic levels.

Despite the diversity of marker types for phylogenomics, it remains unclear whether features specific to each marker type can ultimately result in phylogenomic datasets that can strongly mislead. For example, incongruence in the phylogeny of modern birds developed by Jarvis et al. (2014; 48 whole genomes) and Prum et al. (2015; 259 anchored phylogenomics loci, 198 species) has recently been attributed to differences in marker type rather than number of taxa (Reddy et al., 2017). Whereas Jarvis et al. (2014) used primarily noncoding loci because they observed gross incongruence when using coding regions, the loci used by Prum et al. (2015), although nominally focused on broadly “anchored” conserved regions, came primarily from coding regions. Thus, at least one marker type is likely inappropriate when applied across modern birds (ca. 100 Ma). These data type effects can stem from multiple sources. Selection on exons might lead to localized differences in effective population size across the genome, as previous studies have highlighted issues with base composition heterogeneity within exons across taxa (Figuet et al., 2015; Scally et al., 2012). On the other hand, alignment quality of introns and ultraconserved elements can sometimes be less than desired (Edwards, Cloutier & Baker, 2017). Clearly marker effects can potentially have substantial consequences on species tree estimates and need to be further evaluated and compared side-by-side by the phylogenetic community (Shen, Hittinger & Rokas, 2017).

A priori versus a posteriori selection of loci for phylogenomics

In an ideal world, phylogeneticists would have whole and fully annotated genomes of all taxa available, allowing them to select loci for phylogenomics based on the relative merits of different loci. This a posteriori (i.e., after data generation) selection would be carried out after assessing desired phylogenetic and biological properties of a wide array of markers for which the data are already in hand. A posteriori selection of loci for phylogenomics is clearly a long-term goal that will yield greater choice and justification for specific loci. Today, the loci for phylogenomics are selected a priori (i.e., before data generation) based primarily on cost and ease of collection and alignment, disregarding potentially useful regions of the genome. Thus, an attractive aspect of whole-genome sequencing (WGS) for phylogenomics is to have the opportunity to select markers a posteriori once genomes are in hand (e.g., Edwards, Cloutier & Baker, 2017; Fig. 1). WGS is less prone to sequencing biases and also allows for further expansion into different research fields and questions based on the same initial data (Lelieveld et al., 2015). In contrast, a priori marker selection often limits the kinds of questions and methods that researchers can apply and represents a real constraint for phylogenomics and other disciplines.

Figure 1 A posteriori marker selection from whole-genome alignments for phylogenomics and phylogeography.

Whole-genome analysis (A) permits researchers to choose different markers for specific purposes (B–D). By contrast, subsampling methods such as Rad-seq or hybrid capture, which dominate phylogenomics today, usually yield a specific set of markers that the researcher has chosen a priori. The generation of WGA thus greatly increases the use of genomic data in biological research, beyond the initial goals of the researcher producing those data. Here, we show how a hypothetical WGA that includes seven different loci (different colors) for four individuals allows extracting sequence data to generate gene trees (B), identifying SNPs to genotype individuals (C), and measuring copy depth to infer CNVs across genomic regions (D). Ultimately, these different kinds of data can be translated into species tree inferences (B–D). In the case of CNVs, only locus number 3 (orange) shows significant CNV. Because CNVs are measured as continuous characters (i.e., copy depth), the orange shading represents a hypothetical evolutionary scenario of copy number variation of genomic region number 3 within the inferred species tree, which is incongruent with those based on sequence and SNP data from other loci in the genome.

An important constraint for using WGS for downstream phylogenomic analyses is genome quality. Obtaining high-coverage well-assembled and thoroughly annotated genomes is still very expensive and time-consuming, and even low-coverage genomes are still outside reach for large portions of the scientific community. However, even low-coverage genomes, which should be cautiously used for maker selection due to potential problems caused by poor annotation and coverage, can sometimes yield a modest number of markers for phylogenomics, and in the short term might even yield data sets allowing a broader diversity of markers for analysis. Although we are fully aware of these constraints, we are particularly excited about the potential that we see in routinely using WGS to produce phylogenomic data sets.

More taxa versus more loci

The question of whether to add more genes or more taxa was a dominant theme in phylogenetics in the 1990s and early 2000s (e.g., Hillis, 1996; Kim, 1996), and remains a persistent question in phylogenomics today. After much debate in the literature (e.g., Hillis, 1996; Graybeal, 1998; Hillis, 1998; Poe, 1998; Mitchell, Mitter & Regier, 2000), the initial consensus view from the Sanger sequencing era of phylogenetics, is that adding more taxa generally improves phylogenetic analysis more so than more markers (e.g., Hillis, 1996; Graybeal, 1998; Poe, 1998). However, phylogenomics is adding a new twist to this consensus, both from the standpoint of data acquisition and from theory (e.g., Rokas & Carroll, 2005; Nabhan & Sarkar, 2012; Xi et al., 2012; Patel, Kimball & Braun, 2013). Amassing large data sets, both in terms of more taxa and more loci, is still a guiding principle of phylogenomics. But with the ability now to bring together many different types of markers in a single analysis, and to analyze them in ways that were not previously available, the “more taxa vs. more genes” debate is becoming more nuanced (Nabhan & Sarkar, 2012). For example, recent work shows that this debate can be highly context-specific and model-dependent (e.g., Baurain, Brinkmann & Philippe, 2006; Dell Ampio et al., 2013; Edwards et al., 2016b). Also, coalescent methods appear to be more robust to limited taxon sampling than traditional methods like concatenation (Song et al., 2012; Liu, Xi & Davis, 2015). Some researchers favor “horizontal” data matrices, wherein the number of loci far exceeds the number of taxa, whereas other researchers favor “vertical” matrices, where many taxa are analyzed at just a few (1–5) loci. Whereas the PCR era of phylogenetics was often dominated by vertical matrices, HTS is allowing data matrices to become more horizontal (Fig. 2). It is important to note that as these more horizontal data become more prevalent, they increase the amount of missing data and aligning problems that can contribute to misleading or low phylogenetic resolution. At least in a coalescent framework, scaling up in both dimensions will be crucial for improved phylogenies and the number of loci required to resolve a given phylogenetic problem is often a function of the coalescent branch lengths in the phylogenetic tree being resolved, with longer branches requiring fewer loci (Edwards, Liu & Pearl, 2007; Huang et al., 2010). At deeper time scales, increasing the number of loci have not proven particularly useful to resolve problematic areas in the ToL (e.g., sister group to animals; King & Rokas, 2017). Despite methodological complications in the accurate estimation of population genetic parameters and computational limitations for MCMC convergence at deep times in coalescent-based analyses, recalcitrant nodes likely represent true complexities in the diversification history of these groups and not necessarily reflect failures of coalescent-based phylogenetics (Lanier & Knowles, 2015).

Figure 2 Trends in phylogenomic data sets since the emergence of HTS.

Based on a sample of 164 phylogenomic papers published since 2004 (see Table S1), we observed no increase in the number of species per data set over time (A). On the other hand, there is a significant increase in the number of loci (B), total alignment length (C), and total data set size, as measured by the product of species times locus number (Data set size 1, E) and species times total alignment length (Data set size 2, F). Moreover, the advent of HTS does not support the notion of a tradeoff between the number of species and the number of loci in phylogenomic studies (D).

To study how researchers have resolved challenges of balancing numbers of taxa versus numbers of loci, we quantified trends in phylogenomic data set size and structure over the past 13 years, drawing data from 164 data sets across diverse taxa (Table S1). We found that, whereas the number of species per paper has not increased significantly over time (Fig. 2A), there were significant increases with time in number of loci (Fig. 2B), total length of sequence analyzed (Fig. 2C), as well as total data set size, as measured by the product of species times locus number (Fig. 2E) or species times total alignment length (Fig. 2F). These results mirror similar trends evaluated for the size of data sets in phylogeography (Garrick et al., 2015). Surprisingly, we found no evidence for a tradeoff between the number of species investigated and the number of loci analyzed (Fig. 2D); perhaps HTS data sets have plateaued somewhat in terms of number of loci, whereas the number of species analyzed is more a function of the questions being asked and the clade being investigated. Regardless, we suspect that, in general, the number of loci and total alignment lengths in phylogenomic data sets are likely a function of resources and sequencing effort. The era of whole-genome sequencing in phylogenomics is still dawning, given that most studies thus far have used targeted approaches for sampling loci (Table S1). We suspect that once whole-genome sequencing on a clade-wide basis become routine (e.g., Genome 10K Community of Scientists, 2009; Grigoriev et al., 2014; Cheng et al., 2018), we will witness yet another jump in the sizes of phylogenomic data sets.

Filtering heterogeneous phylogenomic data sets

Recent studies show that the addition of more loci and more taxa can result in higher levels of gene-tree discordance (e.g., Smith et al., 2015; Shen, Hittinger & Rokas, 2017). This is not unexpected - as the number of taxa and loci increase, the greater the likelihood of capturing signals of the heterogeneous evolutionary history (e.g., ILS, lateral gene transfer (LGT), hybridization, gene duplication and loss (GDL); See Table 1 for a definition of these concepts), misidentifying orthologs from paralogs, and recovering patterns of molecular evolution (e.g., noise/lack of signal in the sequences, and nonstationarity in base composition) that can contribute to gene tree discord. At the same time, the variance in gene tree topologies could also have been caused by errors in gene tree estimation. Such observations have been used to argue that the accuracy of gene tree inference should be maximized or at least evaluated, but it is not clear what criteria should be used to filter sets of gene trees. For example, filters can be based on rates of molecular evolution (Klopfstein, Massingham & Goldman, 2017), levels of phylogenetic informativeness (Fong et al., 2012), or on the cause of gene-tree discord itself, if known (Huang et al., 2010). Chen, Liang & Zhang (2015) found that selecting genes whose trees contained a well-known uncontested and long branch in a given species phylogeny (long enough so as not to incur substantial ILS) was a better way to improve phylogenomic signal than selecting genes based on characteristics of sequence evolution. All of these methods are excellent suggestions and should be explored further. Still, the effects of such culling on the distribution of gene trees, and whether it could distort the distribution so that it departs from models like the multispecies coalescent, are unknown, and potentially of concern (but see Huang et al., 2017). Aside from the use of some explicit methods for detection of outlier genes (e.g., de Vienne, Ollier & Aguileta, 2012), rogue taxa (e.g., Aberer, Krompas & Stamatakis, 2013), outlier long branches (e.g., Mai & Mirarab, 2018), and tree space visualization (e.g., Huang et al., 2016; Jombart et al., 2017), an obvious way to alleviate potential effects of gene tree outliers is a more balanced taxon sampling (Hedtke, Townsend & Hillis, 2006). Nonetheless, we need further studies on the effects of different types of phylogenomic filters on the properties of large-scale phylogenomic datasets.

Table 1 Definitions of core concepts used in this article.

Concept	Definition	
The Tree of Life (ToL)	This idea, originally articulated by Darwin and others, refers to the grand vision of understanding the branching pattern of all life on earth. Today the idea conveys the use of morphological and molecular data to reconstruct the phylogenetic relationships of all life forms. In some usages, the idea also includes reconstructing reticulate evolutionary events, such as introgression and hybridization, which are now thought to be common in many lineages.	
High-throughput sequencing (HTS)	Also referred to as “next generation sequencing”, this term refers to the plethora of new DNA and RNA sequencing technologies that in the last fifteen years have allowed biologists to dramatically increase the number of bases sequenced for a given species or clade. HTS technologies can be applied to sequencing whole genomes or transcriptomes and have been embraced by phylogeneticists interested in increasing the size of comparative molecular data sets. See Goodwin, McPherson & McCombie (2016) for a review on the progress of HTS.	
The multispecies coalescent model (MSC)	A generalization of the standard, single population coalescent model to multiple species related in a phylogeny. The MSC applies the single-population coalescent model to each branch of a phylogenetic tree, including both terminal and internal branches. In the MSC, alleles sampled in terminal species will coalesce to a smaller number of ancestral alleles at a rate depending on the effective population size within the branch. The gene tree lineages in a branch of the species tree do not necessarily coalesce within that branch as one goes backwards in time; multiple alleles may persist into ancestral branches. This phenomenon is called incomplete lineage sorting (see next definition). The decrease in the number of alleles and the time to coalescence to a single allele in a lineage follows the standard neutral coalescent model, until all alleles coalesce from all species. See Rannala & Yang (2003) and Degnan & Rosenberg (2009) for a full discussion.	
Incomplete lineage sorting (ILS)	This phenomenon, originally described by John Avise (see Avise et al. 1987) refers to the tendency of alleles in an ancestral species to persist across multiple speciation events, resulting in a situation in which the gene tree (“allele tree”) differs from the species tree. In ILS, alleles fail to “sort” by genetic drift as species diverge from one another, resulting in different species retaining the same alleles, or their descendants, causing discordance with the overarching species or population tree. The language of this phrase is looking forward in time, as opposed to the language of coalescence, which looks backwards in time. See Degnan & Rosenberg (2009) for a full discussion.	
Gene duplication and loss (GDL)	This concept describes the process by which a gene in an ancestral species can duplicate, forming paralogs and one or more of the paralogs can subsequently be deleted from the genome, resulting in complex patterns of relationships among paralogs and orthologs. Gene duplication is another mechanism, in addition to ILS, that can render the gene tree different from the species tree. As a result of gene (paralog) loss, inferring the correct ortholog/paralog relationships and history of branching events in a multigene family can be challenging. Phylogenetic models incorporating GDL try to use patterns in multigene families to deduce the branching history of the constituent species. See Degnan & Rosenberg (2009) and Sousa et al. (2017) for a full discussion.	
Ancestral recombination graph (ARG)	This is a complete record of the coalescent and recombination events in the history of a set of DNA sequences. As a consequence of incorporating recombination events, ARGs do not necessarily depict trees, but often have a network structure. The accurate estimation of ARGs remains challenging but they enhance our ability to estimate recombination rates, ancestral effective population sizes, population divergence times, rates of gene flow between populations, and detect selective sweeps. See Griffiths & Marjoram (1996), Siepel (2009), and Rasmussen et al. (2014) for a full discussion.	
Lateral gene transfer (LGT)	This process occurs when genes jump taxonomic and phylogenetic boundaries, moving between unrelated species and therefore causing discordances between genetic and lineage history. LGT, along with ILS and GDL was among the three primary causes of discordance between gene and species trees identified by Maddison (1997). LGT has been documented to occur between bacterial lineages, between bacteria and viruses, and between these two and eukaryotes, including plants and vertebrates. If not identified prior to phylogenetic analysis, LGT can cause many algorithms for phylogenetic inference to fail. Without prior identification, LGT essentially amounts to errors in data sets and sequence alignments. At the same time, LGT can be a source of adaptation and evolutionary novelty for recipient genomes and has had a major impact on the history of life. See Gogarten & Townsend (2005) for a full review.	

High-throughput sequencing opens possibilities for new information and marker types

Heterozygosity and intra-individual site polymorphisms

Some of the prevalent occurrences in organisms with multiple ploidies are intra-individual polymorphisms and increased heterozygosity. However, due to issues such as lack of sufficient read coverage and connectivity, confident identification of such polymorphism continues to be challenging (Garrick, Sunnucks & Dyer, 2010; Lischer, Excoffier & Heckel, 2014; Schrempf et al., 2016) and many data sets do not permit statistical approaches, such as PHASE (Stephens, Smith & Donnelly, 2001), to robustly determine haplotypes of different alleles (Garrick, Sunnucks & Dyer, 2010). Consequently, in phylogenetics, heterozygosity and intra-specific polymorphic sites are often accommodated using UIPAC ambiguity codes or ignored entirely or by randomly selecting alleles (Iqbal et al., 2012). In fact, most “one sequence per individual/species” phylogenomic data sets consists of haplotypes that might not occur in nature because many methods, including de novo assemblies of genomes, yield single haplotypes consisting of consensus or other haplotype summaries from diploid organisms. The fact that HTS produces several reads of the same region allows for the identification of heterozygosity and intra-specific polymorphic sites represents an untapped opportunity to incorporate intra-individual variation in our phylogenetic estimates (Lischer, Excoffier & Heckel, 2014; Schrempf et al., 2016; Andermann et al., 2018). Recent models have been proposed to improve calling and sorting such polymorphisms (De Maio, Schlötterer & Kosiol, 2013; Lischer, Excoffier & Heckel, 2014; Potts, Hedderson & Grimm, 2014; Schrempf et al., 2016) and, although results of different studies vary (Kubatko, Gibbs & Bloomquist, 2011; Lischer, Excoffier & Heckel, 2014), estimation of individual, naturally occurring haplotypes has been shown to improve phylogenomic reconstructions based on genome-scale data (Andermann et al., 2018).

Rare genomic changes

As noted above, molecular phylogenetics has primarily used alignments of sequence-level data for phylogenetic inference. This bias is perhaps driven by the notion that genome evolution occurs by aggregating small changes, such as point substitutions, over time. However, this bias also responds to the challenges of characterizing rare genomic changes, such as indels, transpositions, inversions, and other large-scale genomic events (Rokas & Holland, 2000; Boore, 2006; Bleidorn, 2017). This emphasis on sequence data has produced a vast ecosystem of algorithms tailored to analyze such data, but most phylogeneticists would agree that rare genomic changes would be a welcome addition to the toolkit of phylogenomics, because they are generally regarded as highly informative markers, providing strong evidence of homology and monophyly (Boore, 2006; Rogozin et al., 2008). With the increased availability and affordability of WGS, our view of genome plasticity has changed drastically in recent years and we are now capable of exploring other genomic features beyond the signals encapsulated in DNA or amino acid sequences (e.g., Ryan et al., 2013). The question then arises of how to identify and utilize these rare genomic markers, as well and assess their phylogenetic informativeness (Rokas & Holland, 2000; King & Rokas, 2017). Genome-level characters will likely have different evolutionary properties than sequence-based markers, suggesting that one of the biggest challenges we face for incorporating genomic changes into phylogenetic analyses is to find informative evolutionary models and tools suited for these kinds of data and assess how congruent or discordant they are with respect to other markers (e.g., Rota-Stabelli et al., 2011). This will not only shed light on how phylogenetically informative different genomic changes are, but also will broaden our understanding of the evolutionary intricacies across different genomic regions (Rokas, 2011; Leigh et al., 2011).

Gene order and synteny

Computational algorithms to use gene order and rearrangements as markers in phylogenetics (Tang et al., 2004; Ghiurcuta & Moret, 2014; Kowada et al., 2016) were spurred in part by the seminal paper by Boore, Daehler & Brown (1999) using mitochondrial gene rearrangements to understand the phylogeny of arthropods. Initially, algorithms for making use of gene order and synteny were applied primarily to microbial genomes, but recent efforts have extended such methods to the analysis of eukaryotes as well (see Lin et al., 2013). Gene order and synteny appear most promising at high phylogenetic levels, although we still do not know how informative gene order will be at many levels. For instance, chromosomal rearrangements appear highly dynamic in some groups, such as mammals, and further study of their use in phylogenomics is warranted (Murphy et al., 2005).

Indels and transpositions

Indels and transpositions are two types of molecular characters that are underutilized in phylogenomics. The former perhaps because standard methods of analysis often treat indels as missing data and the latter because they are technically challenging to collect without whole-genome data. Indels have been used sporadically in phylogenomics and several researchers have argued for their utility and informativeness, given appropriate analytical tools (Jarvis et al., 2014; Ashkenazy et al., 2014; Roncal et al., 2016). Murphy et al. (2007) used indels in protein-coding regions to bolster estimates of mammalian phylogeny and found that the Atlantogenata hypothesis was supported after scrutinizing proteome-wide indels for spurious alignments and orthology. The Avian Phylogenomics Project (Zhang, Jarvis & Gilbert, 2014) found that indels had less homoplasy than SNPs and, despite showing high levels of ILS, was largely congruent with other markers across the avian tree. Transposable elements arguably are even more highly favored by phylogenomics researchers, but are much more difficult to isolate and analyze and have been used principally across various studies in mammals and birds (Kaiser, van Tuinen & Ellegren, 2007; Churakov et al., 2010; Kriegs et al., 2010; Suh et al., 2011; Baker et al., 2014; Cloutier et al., 2018a). Whereas they are generally considered to have a low rate of homoplasy, most researchers agree that they can in some circumstances exhibit insertional homoplasy. Moreover, no marker is immune to the challenges of ILS, and transposable elements and indels are no exception (Matzke et al., 2012; Suh, Smeds & Ellegren, 2015). Still, the exceptional resolution afforded by some studies employing transposable elements is exciting, and we expect this marker type to increase in use as whole genomes are collected with higher frequency.

Copy number variations

The 1000 Genomes Project estimates that in humans about 20 million base pairs are affected by structural variants, including copy number variations (CNV) and large deletions (1000 Genomes Project Consortium, 2015), suggesting that these types of mutations encompass a higher fraction of the human genome than do SNPs. A CNV is a DNA segment of at least one kilobase (kb) that varies in copy number compared with a reference genome (Redon et al., 2006). CNVs appear as deletions, insertions, duplications, and complex multi-site variants (Fredman et al., 2004). Such a profusion of CNVs across human genomes has proven useful in tracking population structure (Sjödin & Jakobsson, 2012), but still remains underappreciated in phylogenetics.

Newly available methods allow inference of CNV at high resolution with great accuracy (Wiedenhoeft, Brugel & Schliep, 2016). The frequency with which CNVs occur in animal and plant populations raises the question of how informative they would be at higher phylogenetic levels, and whether they would incur unwanted homoplasy that would obscure homology and phylogenetic relationships. For example, some CNVs evolve so quickly that they can be used with success at the sub-individual level, for example, in tracking clonal evolution of cancer cells using CNV-specific phylogenetic methods (e.g., Schwartz & Schäffer, 2017; Liang, Liao & Zhu, 2017; Ricketts et al., 2018; Urrutia et al., 2018). Moreover, their interspecific variation has been shown to correlate with the phylogeny of some groups, such as the highly pathogenic and rapidly-evolving barley powdery mildews (Frantzeskakis et al., 2018). However, such fast evolution may mean that these markers might be less useful at higher levels of biological organization. Additionally, the adaptive nature of CNVs may or may not facilitate clear phylogenetic signals. For example, a study in Arabidopsis thaliana showed that adaptation to novel environments, or to varying temperatures, is associated with mutations in CNVs (DeBolt, 2010). If CNVs are to become a useful tool in phylogenomics or phylogeography, we must understand their microevolutionary properties in greater detail. For example, the pattern of evolution of CNVs, wherein deletions of genetic material may not easily revert, resulting in a type of Dollo evolution, might help clarify the overall structure of the models applied to them (Rogozin et al., 2006; Gusfield, 2015).

Recent advances in the generation of high-throughput sequence data and their impact on the reconstruction of the Tree of Life

As sequencing technology rapidly moves forward (reviewed by Goodwin, McPherson & McCombie, 2016), our ability to accurately identify the aforementioned marker types increases considerably. For instance, the low per-base error rate of short-read sequencing technologies, such as the Illumina HiSeq X Ten and NovaSeq (Illumina, San Diego, CA, USA), allow for a significant reduction in the cost of sequencing which can result in data for more taxa at a higher coverage. This is certainly beneficial for the accurate identification of SNPs, heterozygosity, and intra-individual site polymorphisms (Goodwin, McPherson & McCombie, 2016) and their use in a phylogenomic context. Moreover, single molecule real-time sequencing technologies, such as Pacific Biosciences (Pacific Biosciences of California, Menlo Park, CA, USA) and Oxford Nanopore (Oxford Nanopore Technologies, Oxford, UK) produce reads that exceed 10 kb (Rhoads & Au, 2015; Lu, Giordano & Ning, 2016). The advent of these technologies has led to improved and more efficient assembly methods that allow accurate identification of structural changes (e.g., Khost, Eickbush & Larracuente, 2017; Merker et al., 2018). When combined with data resulting from short-read sequencing, they represent a powerful tool to correctly identify and use a wide array of genomic markers for numerous purposes. These technical advances, which include portable devices that can be carried into the field (i.e., Oxford Nanopore; Johnson et al., 2017), will certainly yield an increase in the genomic loci and taxa available for genomic and phylogenomic studies.

Concepts and Models in Phylogenomics

For decades, phylogenetics has struggled with how best to translate evolutionary changes in DNA sequences and other characters into phylogenies, and genomic data are no exception to this trend. Phylogenomics is still in a developing stage of formulating models that effectively represent the underlying mechanisms for genome-scale variation while remaining efficient and within reasonable analytical and bioinformatic capacities. The current focus on models and evolutionary forces generating the patterns that we recover as branching and reticulation events in our phylogenetic reconstructions is a healthy one and can be extended to other important topics in phylogenomics, such as species delimitation, character mapping, and trait evolution (e.g., Yang & Rannala, 2014). All of these areas are developing rapidly and are in need of updated models and bioinformatics applications to cope with the heterogeneity brought by genome-scale data.

The multispecies coalescent model

One of the key practical advances in molecular phylogenetics has been the incorporation of gene tree stochasticity into the inference of species phylogenies, via the multispecies coalescent model (MSC: Rannala & Yang, 2003; Liu & Pearl, 2007; Heled & Drummond, 2010). The MSC allows gene trees to be inferred with their own histories, including coalescent-appropriate branching models, but contained within independent yet connected lineages within a species phylogeny, with speciation-appropriate branching models (Degnan & Rosenberg, 2009). The main conceptual advance has been to understand and separately manage the variation at different levels of biological organization—an advance that began years ago (Doyle, 1992; Maddison, 1997; Pamilo & Nei, 1988), but has only recently been widely embraced and put into practice (Edwards, 2009b). Given its ability to accommodate heterogeneous histories across loci scattered throughout the genome, the MSC lays at the core of the conceptual framework to deal with genome-scale data (e.g., Rannala & Yang, 2008; Liu et al., 2015). In the few instances in which model comparison and fit has been evaluated (Liu & Pearl, 2007; Edwards, Liu & Pearl, 2007), the MSC vastly outperforms concatenation. This of course does not mean that the MSC is the correct, or even an adequate, model for phylogenomic data (Reid et al., 2014). Despite concerns regarding some of its implementations when dealing with genomic data (e.g., Springer & Gatesy, 2016), the MSC is a powerful theoretical model for phylogenomics and there is room for refinement and improvement for its applications (e.g., Edwards et al., 2016b, Xu & Yang, 2016).

Bypassing full likelihood models by relying on summaries of the coalescent process

Given the huge computational resources required for modelling all the complexities of evolutionary processes in a statistical framework, there is interest in methods that will accommodate genome-scale data for large numbers of species within a coalescent framework. The utility of such methods cannot be overstated: the rapid rise of large-scale genomic data sets has clearly outstripped theoretical and computational methods required to analyze them. For example, although progress is being made regarding scalability of full Bayesian methods of species phylogeny inference (e.g., Ogilvie, Bouckaert & Drummond, 2017), they are still unable to accommodate large phylogenomic datasets, which often consist of hundreds of species for thousands of loci (Table S1). A common approach to speeding up species phylogeny inference consists of ‘two-step’ methods, wherein gene trees are estimated first and separately from the species phylogeny; then, using various summaries of the coalescent process for collections of gene trees, a species phylogeny is estimated. Many useful methods for estimating species phylogenies in this way have been proposed (see Marcussen et al., 2014; Liu, Wu & Yu, 2015; Mirarab & Warnow, 2015; Mirarab, Bayzid & Warnow, 2016), taking advantage of summary statistics of the coalescent process, such as the average ranks of pairs of species in the collection of gene trees (e.g., STAR: Liu et al., 2009; ASTRAL-II: Mirarab & Warnow, 2015) or the distribution of gene trees containing triplets of species (e.g., MP-EST; Liu, Yu & Edwards, 2010). Some of these two-step methods, while approximate, nonetheless allow for statistical testing in a likelihood framework. For example, MP-EST can evaluate the (pseudo)likelihood of two proposed species phylogenies given a collection of gene trees and the difference in likelihood can be used to evaluate two proposed species phylogenies against each other. However, such statistical approaches have rarely been used thus far, and bootstrapping or approximate posterior probabilities on branches are by far the most common statistics applied to species phylogenies (Sayyari & Mirarab, 2016). Speeding up the estimation process using two-step methods can be effective, but it can also accumulate errors or misallocate sources of variance which cannot be corrected at later stages (Xu & Yang, 2016). If gene trees are biased or uninformative, then downstream analyses for species phylogeny estimation or species delimitation may similarly be compromised (e.g., Olave, Sola & Knowles, 2014). For example, MP-EST can sometimes perform poorly when PhyML (Guindon et al., 2010) is used to build low-information gene trees because PhyML may produce biased gene trees when the alignments contain very similar sequences (Xi, Liu & Davis, 2015). This may account for the lower performance of MP-EST compared to ASTRAL in some simulation conditions, because ASTRAL resolves input polytomies and zero-length branches in gene trees more appropriately. This difference between MP-EST and ASTRAL is eliminated when RAxML (Stamatakis, 2014) is used to build gene trees (Xi, Liu & Davis, 2015).

Beyond the multispecies coalescent model

Reticulation at multiple levels challenges the standard multispecies coalescent model

The phylogenetic processes of branching and reticulation can operate at several levels of organization, including within genes, within genomes, and within populations or species (Figs. 3–5). For example, recombination can cause reticulations within genes, allopolyploidization can cause reticulations at the level of whole genomes, and introgression and hybridization can cause reticulations at the level of populations. These levels are nested so that branching processes (and in part reticulations) acting at a higher level will cause correlated branching patterns at lower levels. At the same time, reticulations at lower levels, such as recombination acting within genes, will cause inference problems at higher levels, such as estimating population histories. Crucially, however, it is only recombination that will break one key element driving many recent models of phylogenetics and population histories, namely dichotomous gene trees. Reticulations at levels of organization higher than the genome, such as the fusing of populations, as well as gene duplication, will still yield collections of dichotomous gene trees, even if the higher-level history is reticulated. Ultimately, the additive effects of these reticulate processes result in our observed phylogenetic reconstructions, and we expect all of these scenarios to produce bifurcating, dichotomous gene trees. From a modelling point of view, another key distinction is whether at the species level, we still have a phylogeny that is tree-like, or whether a network is needed. The process whereby two populations jointly produce a third requires a network to model properly. Allopolyploidy is another situation requiring a network. There are several statistical methods for inferring homoploid networks (Yu et al., 2014; Solís-Lemus & Ané, 2016; Wen, Yu & Nakhleh, 2016; Wen & Nakhleh, 2018), species histories under allopolyploidy (Jones, Sagitov & Oxelman, 2013), and some two-step methods such as PADRE (Huber et al., 2006; Lott et al., 2009). In general, dealing with multiple simultaneous violations of the MSC, such as introgression and allopolyploidy, remains challenging (Degnan, 2018). It is likely that the history of many radiations involves parts of the genome with a dichotomous history and parts that exhibit reticulation, demanding methods that accommodate both scenarios. Alternatively, rather than trying to accommodate multiple processes in our methods for phylogenetic inference, we might instead focus our attention on subsets of loci that would not violate the MSC (e.g., Knowles, Smith & Sukumaran, 2018). In cases where processes other than ILS contribute to gene tree discord (i.e., the distribution of trees is statistically inconsistent with expectations under the MSC; see Smith et al., 2015), loci consistent with the MSC can be identified (e.g., separated from loci with horizontal gene transfer) using CLASSIPHY (Huang et al., 2017). It has also been suggested that in order to distinguish violations of the MSC, the MSC can be used as a null model to be compared with increasingly complex models that would invoke processes such as hybridization and recombination using networks (Degnan, 2018). To follow this promising approach, further research must be conducted to not only model specific processes, but also distinguish them.

Figure 3 Some examples of violations of the multispecies coalescent.

In event A, there is gene flow; in event B there is homoploid hybridization; in event C, there is a gene duplication; and in event D, incomplete lineage sorting. All of these processes contribute to gene tree heterogeneity but fall outside the standard multispecies coalescent model. Importantly, all of these processes also yield strictly dichotomous gene trees, whereas recombination (not illustrated here) does not.

Figure 4 Gene duplication and loss (GDL) creates patterns that can mimic incomplete lineage sorting and other processes, leading to spurious inferences of the species history.

Genes and genomes of three species A, B, and C. Multi-colored bars show (parts of) their genomes with a number of loci indicated in different colors. The orange gene is duplicated in species A and it was lost in species B. The blue gene was duplicated before the divergence between species A and the ancestor of species B and C. However, one of these copies was lost in species A, whereas both copies were maintained in species B and C. Reconstruction of the orange gene tree based on extant diversity will yield a wrong inference of its history due to the absence of data for species B. On the other hand, a phylogenetic reconstruction of the blue gene is difficult to predict. Depending on which of the duplicates are sampled for species B and C, different outcomes can be expected regarding the relationship among the three species. The duplication and loss history of these two genes may cause serious issues for phylogenetic reconstruction because no specific pattern can be expected between them.

Figure 5 Complex patterns of gene lineages with polyploidization and interspecific gene flow.

Genes and genomes of four species A, B, C and D. Multi-colored bars show (parts of) genomes with a number of loci indicated in different colors. Two gene trees, one orange and one blue, evolve within the species network. Species B is an allopolyploid containing two genomes.

Models accommodating dichotomous divergence with gene flow are somewhat limited. For example, in IMa2 (Hey & Nielsen, 2004; Hey & Nielsen, 2007; Hey, 2010) the species phylogeny must be known and fairly small; in the method of Dalquen, Zhu & Yang (2017), both the species phylogeny and gene trees are restricted to three tips. Looking forward, it may be useful to deal with two sub-problems: The first is estimating the species phylogeny despite migration, for example by identifying which loci are interfering with the species phylogeny inference or causing reticulations in the form of gene flow. The second sub-problem is to incorporate a gradual speciation process (Fig. 6), where gene flow after speciation slowly declines, perhaps according to some simple function like an exponential. Such a model would capture what is thought to be a more common speciation process than the instantaneous process modelled by the MSC (Jones, 2018).

Figure 6 Gradual speciation, or isolation-with migration.

After starting to split, gene flow between species decreases gradually. Such a gradual decrease in the extent of gene flow between species might present an especially useful extension of the standard multispecies coalescent model. Colors depict different gene pools and their gradual change along branches describes how species gradually differentiate despite the existence of migration over time. Thickness and color intensity of arrows show that gene flow becomes weaker as species gradually isolate.

In some cases, it is possible to model one situation with either a species network or a tree with gene flow. Long (1991) discussed two models of admixture: Intermixture and gene flow (Fig. 7). The phylogenetics community has mainly focused on methods for inference under the intermixture model (e.g., the multispecies network coalescent; Yu et al., 2014, whereas the population genetics community has focused more on models including gene flow (e.g., IM (Hey & Nielsen, 2007), G-PhoCS (Gronau et al., 2011), PHRAPL (Jackson et al., 2017), admixture graphs)). While some initial work to test inference based on one of these models on data generated by the other has recently appeared (Wen & Nakhleh, 2018; Solís-Lemus, Bastide & Ané, 2017; Blischak et al., 2018; Zhang et al., 2018), much more work is needed to bring together these two lines of work. Simulations and comparisons of observed and expected summary statistics, such as the site-frequency spectrum (Excoffier et al., 2013), have proven especially useful in distinguishing such scenarios (Fig. 7).

Figure 7 Two possible species phylogenies producing similar observations at present time.

(A) species tree with gene flow. (B) Species network with homoploid hybridization. Distinguishing two such scenarios usually requires simulations and comparison of observed and expected summary statistics.

Reticulation in the form of gene flow or introgression is probably the most difficult violation of the MSC to address (but see Hibbins & Hahn, 2018 for a model that estimates the timing and direction of introgression based on the multispecies network coalescent), in part because the number of potential trees accommodating a reticulating network is even higher than the already high number of trees for a given number of taxa. There is at least one issue where reticulation presents an opportunity as well as a challenge. Any kind of gene flow/hybridization means that there is the possibility of inferring the existence of extinct species, because extinct species contribute novel alleles that exceed the coalescence time of most alleles in the focal species under study (Hammer et al., 2011). Well-known examples are the documented presence of Neanderthal genes in most human genomes due to introgression (e.g., Meyer et al., 2012) and the presence of genomes derived from now-extinct diploids in extant allopolyploids (i.e. meso-allopolyploids; e.g., Mandáková et al., 2010; Marcussen et al., 2015). Some current models can explain the data as containing genetic information from extinct species, but they do not model the full species phylogeny: such a generalized approach seems a promising avenue to explore.

Polyploidy and the challenges of analyzing gene duplication and loss

The MSC model describes well allelic lineages and the mutations they accumulate (Fig. 3; Degnan & Rosenberg, 2009; Liu, Xi & Davis, 2015). The simple MSC model is challenging to apply to evolutionary events in which the evolving entities (genes or paralogs) duplicate and occasionally go extinct during the evolutionary history of the populations/species and thus cannot be sampled in contemporary population or species. Estimating the existence and number of these “ghost” lineages remains challenging. For example, how can we detect duplication events if one of the duplicated loci is lost in descendant lineages? In the case of polyploidy, two (or more) genomes having separate evolutionary histories end up together in a single individual. What consequences for evolutionary history do genomic conflicts and dosage variation in gene expression impose? Polyploidy also raises technical issues, such as whether or not homoeologous sequences are recovered in standard genomic surveys.

The complication that GDL brings to the inference of species phylogenies has long been recognized (Fitch, 1970). It is therefore surprising that practical solutions to the problem of GDL are almost non-existent, with empirical examples usually based on ad hoc methods and deductions. Ancient duplications where most additional copies are retained in descendent species can be fairly easy to diagnose based on phylogeny (Oxelman et al., 2004; Pfeil et al., 2004). However, resolving duplications becomes more difficult when copy number changes quickly (Ashfield et al., 2012), or when duplications are recent and copy loss is complete or nearly so, thus returning the locus to a single-copy state (Ramadugu et al., 2013). In the latter case, the phylogenetic pattern can mimic that of ILS and become indistinguishable from it (Sousa et al., 2017), generally leaving no trace at all of the loss.

Why is GDL so challenging to implement in theory? The topological and coalescent-time similarities between ILS and GDL complicates extending the MSC to include both processes, unless copy number exceeds one in at least some samples (Fig. 4). Assuming that allelic and homoeologous variation is not confused with the copy number of independently duplicated genes, at the very least, duplicated genes could be handled as independent loci with missing data for some samples with MSC inference. When copy loss is complete, or when the duplication is so recent so as to conflate allelic versus copy variation, these GDL loci have little effect on species phylogeny inference and divergence times, especially if the algorithms used employ averages over coalescence times or other parameters across many gene trees (Liu et al., 2009; Sousa et al., 2017). At high proportions, though, they may cause serious issues for phylogenetic reconstruction, because the unexpected positions of gene duplications in a species phylogeny, coupled with random copy loss, means that no specific pattern is expected among the affected gene trees (Fig. 4). This scenario contrasts with the retention of ancestral polymorphisms, where we know that branches in short species phylogenies (in coalescent units) are the cause (Rosenberg & Nordborg, 2002). Thus, we expect deeply coalescing lineages to occur in specific parts of a species phylogeny with a limited number of topological outcomes and branch lengths limited by effective population size, which is not the case for duplicated genes. A recent approach to identifying genes that are single copy, but have nonetheless been affected by GDL, was made using the genomic location of the loci (Sousa et al., 2017), and could prove useful for distinguishing GDL and ILS.

Recombination

All existing methods for coalescent estimation of species trees and networks make two important assumptions, namely that (1) there is free recombination between loci, and (2) there is no recombination within a locus. These two assumptions address a key concept distinguishing MSC models from concatenation or supermatrix models: it is the conditional independence of loci, mediated by recombination between loci, and not the ability to address ILS or discordance among genes per se. Moving forward, three important questions to address are: (1) How robust are methods to the presence of recombination within loci and/or to the violation of independence among loci? (2) How should we model recombination within the species phylogeny inference framework? and (3) How do we detect it and differentiate recombination-free loci?

Researchers have started to examine the first question and found a detectable effect of recombination only under extreme levels of ILS and gene tree heterogeneity (e.g., Lanier & Knowles, 2012). However, more analyses and studies are still needed to explore a wider range of factors and parameters that could affect phylogenetic inference when the assumption of recombination-free loci is violated (e.g., Li et al., 2018). For answering the second question, one approach involves combining the multispecies coalescent with hidden Markov models (e.g., Hobolth et al., 2007). These methods suffer from the “state explosion problem”, where individual states are needed for the different coalescent histories, and they increase rapidly with the number of taxa in the dataset, making them infeasible except for very small (~4 taxa) datasets. New methods that scale to larger datasets are needed if such approaches are to be useful in practice. A different direction is to devise novel methods for inferring species phylogeny while assuming that the genealogies of the individual loci could take the form of an ancestral recombination graph (ARG: Griffiths & Marjoram, 1996; Siepel, 2009; Rasmussen et al., 2014).

Extending these approaches to address recombination would require the development of new models that significantly extend the multispecies coalescent to account for ARGs within the branches of a species phylogeny. For two-step species tree methods, this entails developing new methods that infer ARGs for the individual loci and methods that infer species phylogenies from collections of ARGs. For single-step Bayesian methods, novel developments are needed to sample species phylogenies, locus-specific ARGs, and their related parameters. It will also be important to better understand the conditions under which ignoring recombination will still yield reasonable estimates of phylogeny. Extending the theory to accommodate ARGs may be of intrinsic interest, but if the parameter space in which recombination is relevant is very small, then practitioners may be able to ignore recombination.

Species concepts and delimitation

Coalescent methods have played an important role in the development and critical evaluation of species delimitation methods because they provide hypotheses for species boundaries based on genetic data and be integrated with phenotypic data (e.g., Solís-Lemus, Knowles & Ané, 2015). Irrespective of traditional species concepts, it is essential that the entities at the tips of the species tree do not violate the assumptions of the MSC, wherein species are defined mathematically (e.g., Rannala & Yang, 2003, Degnan & Rosenberg, 2009): the branches of the species tree constitute species or populations that do not exchange genes. However, the MSC model also carries strict assumptions about the divergence process if the delimited units are to be interpreted as species. Specifically, it is important to emphasize that in the “standard” MSC model, these species represent populations that, immediately after divergence, no longer experience gene flow. Therefore, the species of the MSC model do not necessarily correspond with species as a taxonomic rank, defined by traditional species concepts (Heled & Drummond, 2010): “MSC” species could simply be populations by other criteria, so long as they have ceased to exchange genes, even for a short period of time. In other words, a species tree built under the MSC might then be interpreted as a depiction of the history of the barriers to gene flow among diverging structured populations and this may be particularly true when there is dense spatial and genomic sampling across individuals (Sukumaran & Knowles, 2017). Therefore, in those species tree methods requiring a priori assignments of individuals to species, such assignments may strongly influence the inferred species phylogeny, in the same way that hybridization will have serious consequences on an estimated species phylogeny (Leaché et al., 2014).

Recently, several MSC-based methods that have the ability to simultaneously perform species delimitation and estimate the species phylogenies have been developed and implemented (e.g., Yang & Rannala, 2014; Jones, Aydin & Oxelman, 2015; Jones, 2017). These methods seem to consistently recover the correct number of “MSC species” given the assumptions of the model. However, it is probable that the assumption of no gene flow between the descendant populations is often violated and that most reproductive isolation processes are gradual or episodic rather than sudden and permanent (e.g., Rosindell et al., 2010). There is thus need for methods that perform simultaneous species phylogeny estimation and assignment of individuals to species while considering the limitations of the MSC (Jones, Aydin & Oxelman, 2015).

If one prefers a species concept that affirms that most recently diverged populations are necessarily reproductively isolated, current methods will overestimate the number of species as defined by traditional species concepts and will likely reveal instead intraspecific population structure (Sukumaran & Knowles, 2017). Toprak et al. (2016) used DISSECT (Jones, Aydin & Oxelman, 2015) but also employed checks as to the integrity of various hypotheses of species boundaries suggested by the data. From a computational point of view, any species delimitation method will need an operational definition of species. Therefore, a possible development of MSC-based species delimitation methods could be allowing migration and assuming that speciation is complete when a certain proportion of the migrations is reached or when the migration rate is sufficiently low. However, this solution will not be suited for the protracted speciation model because other kinds of information besides the movement of genes will still be needed to identify when a clade becomes reproductively isolated. Possibly the best way to avoid confusion is to restrict the word “species” to taxonomy and base it on multiple sources of information which are synthesized in an integrative fashion (Dayrat, 2005; Will, Mishler & Wheeler, 2005; Bacon et al., 2012; Solís-Lemus, Knowles & Ané, 2015), and refer to the reproductively isolated units of MSC analysis as “MSC units” or “MSC taxa”.

Models at the Intersection of Phylogenomics and Macroevolution

At the intersection of phylogenomics and macroevolution, phylogeneticists aim at shedding light on how patterns of organismal diversity have been generated and maintained through time. The former focuses primarily on building phylogenies, whereas the latter uses them to study the tempo and mode of diversification over time. In many important respects, these two sub-disciplines have remained distinct and non-communicative. On the one hand, phylogenomics and phylogeography have not exhaustively aimed to address the type of questions—related to diversification and trait evolution—that macroevolution focuses on. On the other hand, macroevolution ignores many kinds of complexities inherent to the phylogeny building process that phylogenomics has recently begun to address.

Macroevolutionary models focus on long-term processes, in terms of both species richness and phenotypic diversity. They rely on two types of models: birth-death models of diversification aimed at understanding how and why speciation and extinction rates vary through time and across lineages (Hey, 1992; Nee, Mooers & Harvey, 1992; see Stadler, 2013 and Morlon, 2014 for review) and models of trait evolution aimed at understanding the mode and tempo of phenotypic evolution (Felsenstein, 1973; see Pennell & Harmon, 2013 and Manceau, Lambert & Morlon, 2017 for reviews). These models are typically constructed at the level of species, ignoring the populations or individuals that constitute these species (but see Manceau, Lambert & Morlon, 2015 and Rosindell, Harmon & Etienne, 2015 for exceptions). As a consequence, microevolutionary processes, such as coalescence, have informed phylogenetic methods for building phylogenies more so than have macroevolutionary methods that use them. For example, the most widely used phylogenetic dating methods generally do not acknowledge the critical distinction between speciation times, which are usually of primary interest, and coalescence times, which are often assumed to represent speciation times but in fact represent events older than the divergence of the species concerned (Edwards & Beerli, 2000; dos Reis, Donoghue & Yang, 2016; Angelis & dos Reis, 2015). In addition, macroevolutionary models are fit to species phylogenies (diversification models) or a combination of species phylogenies and phenotypic data (trait evolution models), most often assuming that evolution is best represented by a species tree, not a network (but see Jhwueng & O'Meara (2015); Bastide et al. (2018); Solís-Lemus, Bastide & Ané (2017) for models of trait evolution on networks), and that the species phylogeny is known. Nearly all models that use phylogenies to study character evolution assume a single underlying species phylogeny on which characters evolve. But it has become evident recently that different characters in principle have different phylogenies, for the same reason that genes themselves might have different phylogenies (Hahn & Nakhleh, 2016). Analyzing incongruences between character evolution inferred from the species tree versus from gene trees that are more directly linked to the character under study would provide a refined understanding of character evolution. Recent work on the phylogeny of quantitative characters may be helpful in this endeavor (Felsenstein, 2012).

Developing research projects that integrate the heterogeneity inherent in phylogenomics and macroevolution will bring important new insights into the evolutionary process. For example, developing diversification and phenotypic evolution models to be fit to networks rather than dichotomous trees will allow estimates of rates of hybrid speciation and phenotypic evolution as well as a better understanding of factors influencing such rates (Bastide et al., 2018). Embracing genetic heterogeneity and the incongruence between gene trees and species phylogenies when applying macroevolutionary models could help us to better understand how speciation proceeds, and also to analyze the coupling between genetic and phenotypic evolution (e.g., is phenotypic convergence coupled or not with genetic convergence in relevant genes?). Developing macroevolutionary models accounting for within-species heterogeneity linked to biogeography could help us understand how biogeographic structuring influences speciation, extinction, and phenotypic evolution.

More generally, evolutionary biologists have yet to thoroughly explore the type of new questions that we are going to be able to address if we are given genomic data at the tips of all species from a phylogeny. Such data could allow us to gain an integrative understanding of three fundamental aspects of evolution: evolution at the molecular level, at the phenotypic level, and at the clade level, as well as the links among them. Are rates of evolution at these three levels correlated? If so, how? Do features of genomes or of genome evolution, such as quantity of transposable elements, substitution rates, number of gene duplications, influence rates of diversification and phenotypic evolution? Clearly, we are only at the beginning of exploring these new possibilities.

Mapping trait evolution on heterogeneous genomic datasets

Mapping the genomic basis of phenotypic traits is a major trend in evolutionary biology today (Elmer & Meyer, 2011; Hoban et al., 2016). Such mapping can be conducted in the context of populations of a single species or, increasingly, via comparisons of species on a phylogeny (e.g., Hiller et al., 2012; Marcovitz, Jia & Bejerano, 2016). Phylogenetic genome-wide association (“PhyloGWAS”) methods identify genomic features in coding or non-coding DNA that exhibit unusual patterns of evolution on branches concerned with repeated evolution of phenotypes, thereby drawing connections between the genomic and phenotypic levels (Pease et al., 2016). Such phylogenomic mapping usually assumes a single phylogeny, the species phylogeny, as a framework for analysis, and therefore ignores genomic heterogeneity. To make PhyloGWAS mapping most efficient it might be more appropriate to use the local topology in the genome for inference and estimation of ancestral states. Estimating genotype-phenotype associations solely on the species phylogeny might yield misleading results regarding the origin and evolution of phenotypic traits (Hahn & Nakhleh, 2016; Mendes, Hahn & Hahn, 2016; Guerrero & Hahn, 2018). Heterogeneity across gene histories has been traditionally considered as “biological noise” when using comparative genomics to map traits, but of course such heterogeneity is the focus of gene mapping efforts at lower taxonomic levels. A recently proposed application of the MSC for quantitative traits, accounting for genealogical heterogeneity improves downstream estimates of mean trait values, phylogenetic signal, and evolutionary rates of traits (Mendes et al., 2018). Furthermore, genome-wide or gene-specific selective sweeps associated with the evolution of a particular phenotypic trait are a major source of genetic heterogeneity among closely related populations or species, and can be captured using outlier statistics, such as Fst or Dxy (Pease et al., 2016). Such selective sweep mapping of genes with large phenotypic effect can now be accomplished with high resolution and precision in genomically poorly studied organisms (Lamichhaney et al., 2015). Apart from providing valuable knowledge on the genetic basis of trait diversification, such data are providing increasing support to the fact that cases of genetic heterogeneity can be profitably used in the effort to understand and resolve evolutionary history, rather than considering it “biological noise.” Such thinking needs to be incorporated into comparative genomics more frequently (e.g., Mendes et al., 2018).

Tree-free methods of character evolution

We have seen that incorporating phylogenetic heterogeneity is a challenge for macroevolutionary models of character evolution. At the other end of the spectrum are a class of methods (so called “tree-free methods”) that attempt to draw inferences and principles about trait evolution without assuming a particular phylogeny. The common situation when analyzing character or trait data correlated by a phylogeny is to assume a stochastic process for the trait, commonly a variation of the Brownian motion (BM; Felsenstein, 1985) or Ornstein-Uhlenbeck (OU; Hansen, 1997) processes. Then, using the estimated phylogeny and measured trait data for each species, the parameters of various evolutionary processes—trait variation, patterns and rates of change, etc.—are estimated, often using maximum-likelihood or Bayesian approaches (see Pennell & Harmon, 2013 and Manceau, Lambert & Morlon, 2017 for reviews). However, given the various logistical and technical challenges of inferring robust phylogenies, exploring tree-free methods might represent a useful mechanism for guiding the study of character evolution for certain groups.

Tree-free comparative methods work by integrating over the space of trees (under a given branching process model). For example, under a pure birth model and with enough tip measurements, the optimum value of the OU process can be estimated as the sample average (Bartoszek & Sagitov, 2015a). Similar results have now been derived for other models of tree growth that include extinction (Adamczak & Miloś, 2014; 2015; Ané, Ho & Roch, 2017). Similarly, the rate of adaptation under the OU process, often modeled as the stationary variance—the ratio of the squared “rate of evolution” (sigma parameter in the OU model) and twice the “rate of adaptation” (the alpha parameter) can be estimated as the sample variance (Bartoszek & Sagitov, 2015a). Teasing sigma and alpha apart, however, requires a tree. The key parameter of the BM model, the rate of evolution, is similarly estimable directly from the trait sample (Bartoszek & Sagitov, 2015b; Crawford & Suchard, 2013), whereas the root state cannot be consistently estimated without a tree (Ané, 2008; Sagitov & Bartoszek, 2012). In addition to providing tree-free estimators of some model parameters, the studies mentioned above also derived Central Limit Theorems that allow computing confidence intervals around these point estimates as well as the sample sizes needed to obtain reliable estimates.

Extinct and unsampled species

A notable case when phylogenomics and macroevolution do meet is in the treatment of extinct or unsampled species in phylogenetic reconstruction and dating. Despite the avalanche of genomic data for an increasing number of species, we still lack sequence data for most species, making it difficult to place them in a phylogeny. Some researchers (e.g., Jetz et al., 2012; Tonini et al., 2016) have used a combination of consensus trees and current taxonomic classifications to impute the phylogenetic relationships of unsampled species. In this case, polytomies are often resolved by using distributions of branching times obtained from macroevolutionary birth-death models (Kuhn, Mooers & Thomas, 2011). While we appreciate the value of these approaches given the real logistic difficulties researchers face as they attempt to obtain samples from around the globe and that methods are now easily applicable, such approaches have generated well-founded concerns about biases in our inferences (Davies et al., 2012; Rabosky, 2015) and the extent of these biases remain largely unknown (Pennell, FitzJohn & Cornwell, 2016). Nonetheless, there are methods that can objectively alleviate issues arising from using incomplete phylogenies in downstream phylogenetic comparative analyses. For example, recent results using conditioned birth-death processes (e.g., Gernhard, 2008a, 2008b; Sagitov & Bartoszek, 2012) show that under constant rate processes the size of the clade contributes information on the height of the tree and also on the coalescence times. Such results can be used to improve the calibration and node dating of the phylogeny when some species are not sampled. One would expect that ignoring the non-sequenced species would incur a bias resulting in shorter tree heights, because less time is usually required to generate fewer tips. Conditioned branching process models can help alleviate this bias. Also, macroevolutionary birth-death models are used as branching process priors in Bayesian molecular dating. The availability of likelihood expressions for incompletely sampled phylogenies (Stadler, 2009; Stadler & Steel, 2012; Morlon, Parsons & Plotkin, 2011) thus allow to date phylogenies while accounting for the fact that we have observed only a certain fraction of unsampled species.

An important source of genetic data for extinct and unsampled species comes from ancient DNA. Current methods now allow sequencing ancestral DNA and incorporating it into phylogenomic, phylogeographic, and population genetic analyses (reviewed by Leonardi et al., 2017), and even obtaining wholegenomes from various samples dating back up to a few thousand years (e.g., Lynch et al., 2015; Cloutier et al., 2018b). The paleontological and paleobiological records represent another source of extinct species for which genomic data is out of reach. Having the opportunity of integrating the temporal and phylogenetic information of extinct data provided by fossils into phylogenomic estimates of extant diversity is a crucial task toward a complete ToL (reviewed by Hunt & Slater, 2016). Besides the possibility of placing fossil data into phylogenetic trees, a growing field of development is that of time-scaling phylogenies using fossil data. Traditionally, node-dating methods have used fossils to bound the minimum age of nodes at the base of extant clades in which fossils likely belong based on morphological comparisons (reviewed by Ksepka et al., 2011) or estimated fossil ages have been used to post-hoc-scale branch lengths (Sanderson, 2002; Smith & O’Meara, 2012; Bapst, 2014). However, there is an increasing interest in simultaneously performing phylogenetic inference and time-scaling phylogenies (Pyron, 2011; Ronquist et al., 2012; Heath, Huelsenbeck & Stadler, 2014). These methods, known as tip-dating, have the advantage of accommodating phylogenetic uncertainty implemented in a Bayesian model-based framework (Hunt & Slater, 2016) and have been implemented in several groups spanning a wide temporal scale (e.g., Pyron, 2011; Pyron & Burbrink, 2012; Ronquist et al., 2012; Heath, Huelsenbeck & Stadler, 2014; Slater, 2015; Larson-Johnson, 2016; Matzke & Wright, 2016). Lastly, these phylogenies containing both extinct and extant diversity serve as framework to conduct robust diversification and trait evolution studies (Hunt, 2013; Pennell & Harmon, 2013; Pyron, 2015).

Building, Updating and Sustaining the Tree of Life

Scalability challenges

Inferring the phylogeny of all living organisms represents a different challenge than inferring the relationships of just a few terminals; often the scale at which new methods are developed and tested is on this latter scale. For instance, for eukaryotes alone, recent conservative estimates indicate that there are ~8.7 million species on Earth and only 9–14% of them have been formally described (Mora et al., 2011). Furthermore, out of 2.6 million taxa currently represented in the Open Tree of Life (https://tree.opentreeoflife.org; Hinchliff et al., 2015), only ~55,000 were gathered from hard-data phylogenies, whereas phylogenetic affinities of the rest were inferred from current taxonomic classifications (McTavish et al., 2015, 2017). These observations suggest that the vast majority of taxa on Earth still await formal taxonomic description and placement in the ToL (Mora et al., 2011; McTavish et al., 2017). As mentioned above, one common challenge that phylogeneticists encounter is the difficulty in accessing samples from rare, endangered, or extinct taxa, particularly in countries where collecting and exporting is difficult. Recent genomic techniques now allow successful results in obtaining valuable ancient DNA data from museum specimens (e.g., Staats et al., 2013; Hykin, Bi & McGuire, 2015; McCormack, Tsai & Faircloth, 2016; McCormack et al., 2017; Ruane & Austin, 2017), and here, we advocate for routine use of these resources to improve taxon sampling, enhance research in phylogenomics and phylogeography, and increase awareness and usefulness of natural history collections.

Despite the great increase in the generation of genomic data across organisms, we are often limited to fast and efficient but simpler, less realistic phylogenetic methods and assumptions, such as IQ-Tree (Nguyen et al., 2015), PhyloBayes (Lartillot et al., 2013), and ExaML (Kozlov, Aberer & Stamatakis, 2015), to deal with large, heterogeneous datasets. The main downside of these options is that they are not a full coalescent framework and thus far rely on data concatenation. Fully coalescent methods, such as the popular phylogenetic software program *BEAST (Heled & Drummond, 2010), are not capable of dealing with more than a few hundred taxa and some dozen loci at a time for a common analysis, and only recently the release of StarBeast2 allows for the use of thousands of loci for tens of taxa (Ogilvie, Bouckaert & Drummond, 2017). To tackle this problem, we encourage the continuing development of methods that are fully scalable and ideally only increase analytical time linearly rather than exponentially with the number of taxa and loci. Phylogenetic methods should also be fully parallelizable (in order to run natively in computer clusters) and contain checkpoints, i.e., be able to resume the analyses from the latest logged file in case an analysis crashes or the user wishes to evaluate partial results. Another point of possible improvement is in dealing with new sequences to be added to a previously large dataset: should the analysis start from scratch, or could there be substantial time gains by letting those sequences find their placement in the phylogeny ‘on the fly’ (e.g., Siu-Ting et al., 2014).

Large scale phylogenies should ideally be based on the best (or most comprehensive) available datasets in terms of taxonomic and molecular sampling and be constructed from the data itself. However, even supermatrix inference conducted under a single analysis can add bias on tree heights and coalescence times when performed across unbalanced sampled clades (a very common case for species-rich clades or understudied taxa), and therefore affect downstream analyses that rely on these parameters (e.g., biogeography, trait evolution, diversification rates). Computing optimally populated datasets that combine the largest number of taxa and loci simultaneously is a complex mathematical problem, but recent approaches (e.g., SUPERSMART—Antonelli et al., 2017; PyPHLAWD—Smith & Brown, 2018) attempt to overcome it objectively, such as applying the knapsack problem to phylogenetics by packing the optimal choice of species and suitable alignments into a minimally sparse supermatrix.

Community initiatives

Building the ToL is a grand challenge in molecular phylogenetics, and one that cannot be accomplished by a single person or institution’s efforts. Several initiatives have been developed in recent years to coordinate efforts and provide the research community with synthetic information. A prominent project is the Open Tree of Life (https://tree.opentreeoflife.org/; Hinchliff et al., 2015). This project provides a synthesis of previously published phylogenies merged through supertree and other grafting methods. One issue faced by the initiative is that it relies on authors uploading their phylogenetic trees to open data repositories, such as Dryad Data Repository (http://datadryad.org/pages/organization; Vision, 2010) or TreeBase (Sanderson et al., 1994; Piel et al., 2009), which at least until recently only occurred in about 17% of cases (Drew, 2013). Substantial curatorial efforts are also critical to facilitate reusability of deposited trees (McTavish et al., 2015). A different approach was taken by Antonelli et al. (2017), who developed a framework for continuously inferring time-calibrated large phylogenies from raw sequence data deposited in GenBank (Clark et al., 2016) in a multi-step method. Similarly, various tools have been developed to make information contained in the ToL available for the general public (e.g., Rosindell & Harmon, 2012; Harmon et al., 2013).

Mapping the Tree of Life

While progress has been made in mapping species distributions at the large scale aiming for improved conservation practices (e.g., the Map of Life collaborative project; https://mol.org/), most initiatives do not map the tips of phylogenetic trees directly onto geographic space, and therefore are limited by current taxonomic knowledge. As spatial variation in biodiversity results from interactions between evolutionary history and environmental factors, explicit connections between the tips of the ToL and geographic ranges will greatly improve biogeographic inferences (Quintero et al., 2015) and our understanding of biodiversity patterns and future trends. Advances in mapping the ToL through earth history using genomic-based phylogenetic inferences over broad scales and explicit spatial models (e.g., geophylogenies and continuous diffusion models: Kidd, 2010) depend directly on locality data that should be made available in raw and ready-to-use formats. Data sharing policies for associated data, such as geographic coordinates and voucher information, is not well established among journals. We argue that editorial boards should try as best as possible to establish data policies that value and encourage the deposit of geographic data associated to vouchered specimens and other associated information available for future reference.

Best practices for building the Tree of Life

Data must be well curated in databases and publicly available

As we are now in the era of big data in biological sciences, adequate reproducibility must be a fundamental endeavor of biodiversity research. Therefore, data publication in open access repositories represents a powerful tool that not only ensures long-term storage and public availability for future research, but also serves as a vehicle for clarifying intellectual rights and scientific merits (Costello & Wieczorek, 2014). Biocuration, the activity of organizing, representing, and making biological information accessible to both biologists and bioinformaticians, has now become an important consideration in building, updating, and sustaining the ToL (McTavish et al., 2017). The exponential growth in the amount of genomic scale data and the increased dependence on the availability of each other's’ data to answer complex biological questions means that there is a need for improved data management, analysis, and accessibility. GenBank has been the main open access repository for annotated collections of publicly available molecular data. Although the data stored in this database usually lists information such as organism of origin and publication details, the utility of molecular data in this database to answer multiple biological questions, such as biogeographic patterns of biodiversity, is often hampered by lack of associated information such as collection locality (Scotch et al., 2011; Gratton et al., 2017) or attachment to a specific voucher specimen. Moreover, recent surveys have shown that fewer than 20% of phylogenetic studies provide access to phylogenetic data (i.e., alignments and phylogenies) and when they do, critical biological information such as complete taxon names is missing (Drew et al., 2013; Magee, May & Moore, 2014).

We propose two urgent actions to advance this key field. First, authors should be encouraged to submit molecular data that is linked to voucher specimens deposited in recognized scientific collection and museums. Second, authors, journals, and curators should encourage all molecular data submitted to include information such as collection locality and details of voucher specimens. In this regard, other global initiatives such as the International Barcode of Life Project (iBOL; http://www.ibolproject.org) have had great success linking molecular data with morphological and distributional data. When all the data produced or published are curated to high standards and made accessible as soon as available, biological research will be able to process massive amounts of complex data much more quickly.

Submitting sequence and tree data during publication is now routine. However, making available all analytical methods such as software and code used to process and analyze data is less widely employed by the phylogenetic community. Facilities such as TreeBase, Dryad Digital Repository, and Github (https://github.com/) provide a platform for the curated storage of the data and bioinformatic pipelines underlying the scientific literature (see McTavish et al., 2015; 2017). Authors and journals should require all published research to include links to raw data, processed data, and all analytical methods used to produce the results presented. In general, we advocate for following best practices of data management and publication to ensure the quality and utility of phylogenomic data and their associated biological information (Stoltzfus et al., 2012; Drew et al., 2013; Costello & Wieczorek, 2014). In putting together Fig. 2, for example, we found that basic information on a given phylogenomic study, such as the number of species or sequences analyzed, or the total number of base pairs in an alignment, were often not reported or difficult to recover; including such information in easy-to-access tables prior to article acceptance would greatly facilitate meta-analyses and syntheses as the number of studies grows (Table S1).

The need for adequate curation of analytical tools

In the same way that data must be adequately stored and curated, analytical tools must be available for future use and should guarantee proper reproducibility (Wilson et al., 2014; Darriba, Flouri & Stamatakis, 2018; DeBiasse & Ryan, 2018). One of the reasons behind the dramatic increase in the number of phylogeographic and phylogenetic studies during the last 20 years is the proliferation of software and bioinformatic tools to process and analyze these data. Thanks to these new methods, it is now possible to implement a wide array of theoretical models that sustain the fields of phylogenomics and phylogeography. As stated above, genome-wide data have notoriously increased the necessity to expand our analytical models, ultimately leading to a stronger demand for computing resources (Darriba, Flouri & Stamatakis, 2018). Given their key role in phylogenomic research, it is advisable that software development, documentation, and availability follow the best possible practices (e.g., Leprevost et al., 2014; Wilson et al., 2014; Guang et al., 2016; Darriba, Flouri & Stamatakis, 2018; DeBiasse & Ryan, 2018). Having both data and analytical tools adequately stored and accessible to the public not only will ensure high reproducibility of previous studies, but, more importantly, will facilitate continuing the construction of the ToL (McTavish et al., 2017).

All contributions toward building the Tree of Life must be properly recognized

Some current publishing practices in the scientific community may unintentionally represent hurdles toward the ultimate end of collecting and disseminating phylogenetic data on which to build a ToL. For instance, the increasing need in many countries and communities for publishing high-impact papers understandably often discourages researchers from releasing their data until their studies are complete and have passed the peer-review process. This is partially explained by the heavy emphasis of top journals on unusually novel and flashy findings as compared to those studies that represent more modest, but just as critical, advances in the understanding of the phylogenetic relationships of the groups. Similarly, this urge to publish high-impact papers often impedes adequate long-term studies that could potentially generate a wider variety of basic data. With the cultural emphasis on impact and numbers and rates of publication, in practice there is often a penalty for long-term studies. Our current climate often values novel results produced in the short term. Consequently, as a community, we must reach an equilibrium between short- and long-term scientific production in a way that values both, encouraging high impact studies bringing radical reorganizations of the ToL, without hurting lower impact research and the ongoing search for innovation.

Moreover, because building the ToL is a slow and daunting task, it is important that, as a scientific community, all contributors to the process receive proper recognition for their contributions, thereby keeping motivation high and retaining our best talent. Unfortunately, some contributors, both institutions and roles within them, receive less recognition in this grand task than others. For example, field biologists that obtain basic natural history information and specimens used for building the ToL (Suarez & Tutsui, 2004), and the natural history museums that house those specimens, are often not recognized sufficiently. As a community, we have been following a trend in which, perhaps inadvertently, we do not value as much the production of basic biological and natural history data. This can certainly be recognized in our national funding practices, which often do not support basic taxonomic or natural history fieldwork at the expense of flashier end-uses of biological specimens. Specimens are the foundation of most phylogenomic and phylogeographic studies, and we should find standard mechanisms not only to acknowledge, but also to encourage the production of these data in an integrative framework. It is time to strengthen those initiatives aimed at recognizing scientific production beyond citations of peer-reviewed literature (e.g., ORCID; https://orcid.org) by giving also credit to the production and impact of basic biology datasets and collected specimens. Providing credit for depositing and generating data by tracking, for example, number of access and downloads or number of studies using genetic data associated to specimens, could represent a formal recognition of the importance of producing and sharing basic biological data could help bridge the gap between naturalists, taxonomists, empiricists, and mathematicians invested on the study of life history.

It will be exciting to have objective estimates that allow tracking the direct and indirect impact of how these data and samples are being used. We are confident that such initiatives will highlight the importance of continuing field- and museum-based research in various fields of biological research (Buerki & Baker, 2016). Furthermore, such cultural shifts will undoubtedly encourage discerning young minds to embrace basic biological research in their academic endeavors, rather than embracing more lucrative and societally appreciated applied fields.

Conclusions

In this perspective, we have attempted to cover ground in the vast arena of issues facing modern phylogenomics today. In Table 2, we summarize some of the most pressing challenges that the field of phylogenomics is experiencing as we use coalescent-based methods toward building the ToL. We have seen how genome-scale phylogenomics, currently on a strong footing as a result of the MSC, is increasingly improved by models that recognize reticulate processes, such as recombination and introgression. In contrast, macroevolutionary models that use phylogenies have yet to embrace the heterogeneity that currently drives many theoretical innovations in phylogenetic reconstruction itself. We have emphasized the need for the phylogenomics community to embrace high standards of data quality, curation and accessibility in its long-term pursuit of the ToL. Such a grand mission requires value and recognition placed not only on the end products of the process, such as publications and trees, but also on the natural history specimens on which phylogenies are based and which are cared for by the community of natural history museums. Building the ToL will require contributions from all sectors of biological and related sciences—from field biology to theory and everything in between—and robust cyberinfrastructures to integrate these diverse and increasingly massive data streams.

Table 2 Challenges in the fields of coalescent-based phylogenomics and implications for unraveling character evolution and the Tree of Life.

Category	Challenge	Proposed strategy	
Data	Integration and assessment of large amounts of data with heterogeneous phylogenetic signal.	Protocols for marker selection should assess markers’ biological relevance and adequacy for the study organism, given the temporal and spatial scales in question, and not only logistical convenience. A posteriori (after data generation) marker selection from whole-genome alignments can be useful to inform these aspects as well as minimize the effects of missing data and varying data quality. Until then, researchers should attempt a higher standardization of markers to facilitate combinatory analyses.	
To discern true phylogenomic heterogeneity from noise and error as well as to identify violations of the MSC, adequate filtering of large phylogenomic datasets should be conducted based on biological and statistical properties of markers (e.g., analyses of gene-tree outliers and rogue taxa).	
Further research on filtering methods as well as on their impact on phylogenomic estimation is still required.	
Inclusion of additional character types into phylogenomic analyses.	Research efforts focused on the adequate identification and utilization of rare genomic changes other than nucleotide substitutions, such as indels, transpositions, inversions, CNVs, and chromosomal rearrangements. Development of new methods not only to infer phylogenetic hypotheses based on these characters but also to integrate them with more traditional sequence data.	
Phylogenetic inference models and methods	Analyses of genome-scale data for large numbers of species within a coalescent framework.	Continue the development of models and methods that allow simultaneous gene tree and species tree estimation within a Bayesian framework (e.g., Ogilvie, Bouckaert & Drummond, 2017) for increasingly large and complex datasets.	
For the time being, two-step methods, particularly those based on biological models and permitting statistical tests of topologies in a likelihood framework (e.g., Liu, Yu & Edwards, 2010), are useful tools to incorporate coalescent information into species tree inference.	
Detection and incorporation of violations of the MSC into phylogenomic inferences.	Extensions of the MSC should seek the inference of reticulate evolutionary histories (i.e., multispecies network coalescent; Yu et al., 2014; Wen et al., 2016) by simultaneously incorporating violations of the MSC (e.g., Wen & Nakhleh, 2018 (reticulation and ILS); Jones, Sagitov & Oxelman, 2013 (allopolyploidy)). Inference methods dealing with GDL and recombination are of high priority.	
Further development of conceptual approaches aimed at detecting and quantifying different underlying biological processes of phylogenetic history (e.g., Jones, 2018 (ILS and migration), Blischak et al., 2018 (hybridization), Hibbins & Hahn, 2018 (direction and timing of introgression), Sousa et al., 2017 (ILS and GDL), Li et al., 2018 (recombination rates)). As proposed by Degnan (2018), using the MSC as a null model within a model selection approach can be a powerful tool to identify violations of the MSC and to deepen our understanding of the biological consequences of these processes.	
Models that integrate phylogenomics and comparative analyses	Integrating different phylogenetic signals into comparative analyses.	Methods and models should attempt to incorporate gene tree incongruence into macroevolutionary models of character evolution. Similarly, integrative studies aiming at unraveling character evolution at the molecular, phenotypic, and clade levels.	
Understanding the genomic bases of character evolution in species trees vs. gene trees.	Methods that estimate phenotype-genotype associations incorporating heterogeneity across gene trees or that at least take into account differential state probabilities stemming from gene tree discordance (e.g., Guerrero & Hahn, 2018). Similarly, extensions of the MSC for quantitative traits that take into account genealogical heterogeneity represent a promising avenue for research and implementation (e.g., Mendes et al., 2018).	
Best practices for building the ToL	Increasing the number of species represented in the ToL while ensuring reproducibility and encouraging community participation	Natural history museums must be central players for providing and analyzing genome-scale data. Genetic resources and specimen collections are fundamental for allowing the acquisition of data for extinct and poorly accessible species. Open access community initiatives must continue to be relevant repositories of the ToL. Adequate methods for curation of data and analytical tools must continue to be a high priority.	

Supplemental Information

Supplemental Information 1 Table S1: Information on number of species, number of loci, and data set size contained in 164 phylogenomic data sets.

Each row represents a data set included in Fig. 2. For further details on how this table was built, please see section “Compilation of data in Table S1” in the Supplementary Material.

Click here for additional data file.

Supplemental Information 2 Supplementary Materials: The compilation of data contained in Table S1 and in Fig. 2.

Click here for additional data file.

This paper is a product of the ‘Origin of Biodiversity Workshop’ organized by Chalmers University of Technology and the University of Gothenburg, under the auspices of the Gothenburg Centre for Advanced Studies (GoCAS). We are particularly grateful to the GoCAS organizers and facilitators, in particular Karin Hårding, Mattias Marklund, Bernt Wennberg, Sandra Johansson, and Lotta Fernström. We thank Johnathan Clark, Alison Cloutier, Phil Grayson, Kathrin Näpflin, Flavia Termignoni, Jonathan Schmitt, Simon Sin, João Tonini, and Pengcheng Wang for help compiling Table S1. Thomas Couvreur, Tobias Andermann, Prosanta Chakrabarty, Alex Pyron, Claire Morgan, Chris Creevey, and one anonymous reviewer provided useful comments that improved the contents of this manuscript.

Additional Information and Declarations

Competing Interests

Author Contributions

Data Availability

Alexander Schliep and Scott V. Edwards are Academic Editors for PeerJ.

Gustavo A. Bravo analyzed the data, prepared figures and/or tables, authored or reviewed drafts of the paper, approved the final draft.

Alexandre Antonelli analyzed the data, authored or reviewed drafts of the paper, approved the final draft.

Christine D. Bacon analyzed the data, authored or reviewed drafts of the paper, approved the final draft.

Krzysztof Bartoszek analyzed the data, authored or reviewed drafts of the paper, approved the final draft.

Mozes P. K. Blom analyzed the data, authored or reviewed drafts of the paper, approved the final draft.

Stella Huynh analyzed the data, authored or reviewed drafts of the paper, approved the final draft.

Graham Jones analyzed the data, prepared figures and/or tables, authored or reviewed drafts of the paper, approved the final draft.

L. Lacey Knowles analyzed the data, authored or reviewed drafts of the paper, approved the final draft.

Sangeet Lamichhaney analyzed the data, authored or reviewed drafts of the paper, approved the final draft.

Thomas Marcussen analyzed the data, authored or reviewed drafts of the paper, approved the final draft.

Hélène Morlon analyzed the data, authored or reviewed drafts of the paper, approved the final draft.

Luay K. Nakhleh analyzed the data, authored or reviewed drafts of the paper, approved the final draft.

Bengt Oxelman conceived and designed the experiments, analyzed the data, authored or reviewed drafts of the paper, approved the final draft.

Bernard Pfeil analyzed the data, authored or reviewed drafts of the paper, approved the final draft.

Alexander Schliep analyzed the data, authored or reviewed drafts of the paper, approved the final draft.

Niklas Wahlberg analyzed the data, authored or reviewed drafts of the paper, approved the final draft.

Fernanda P. Werneck analyzed the data, authored or reviewed drafts of the paper, approved the final draft.

John Wiedenhoeft analyzed the data, authored or reviewed drafts of the paper, approved the final draft.

Sandi Willows-Munro analyzed the data, authored or reviewed drafts of the paper, approved the final draft.

Scott V. Edwards conceived and designed the experiments, analyzed the data, prepared figures and/or tables, authored or reviewed drafts of the paper, approved the final draft.

The following information was supplied regarding data availability:

The raw data are available in Table S1.

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
