# Peer review of "Embracing heterogeneity: coalescing the Tree of Life and the future of phylogenomics"

_PeerJ, doi:10.7717/peerj.6399_

## Round 0.1 · original submission · Minor Revisions

Overall the reviewers felt that this was a welcome contribution to the field and each have suggested minor edits to address. One thing that I would give serious consideration is refocussing the title and abstract to better reflect the specific area of phylogenomics that this review addresses (i.e. more towards multispecies coalescent models rather than concatenated alignment approaches which some may expect based on the current title) . This has been reflected in the comments of the reviewers also and could help your review better reach its target audience.

·

Basic reporting

Fulfills all standards

Experimental design

Fulfills all standards

Validity of the findings

Fulfills all standards

Additional comments

Resulting from an international Tree of Life workshop, the authors give a “state of the field” address, and summarize current issues, developments, and future avenues for phylogenomics. In some ways, they are simply summarizing what is common knowledge among in-depth practitioners in the field, but I think such a summary will probably serve as a useful introduction for students, and a touchstone citation for those needing to refer to myriad issues that “everyone knows,” but are often lacking from the literature.

There are a few places where I disagree with how the authors characterize certain issues (some are listed below), and I suspect that most readers will feel this way about certain parts. This is probably endemic to a large group effort to summarize an entire field, and isn’t necessarily a condemnation of the authors. Three criticisms in particular:

1) The authors focus heavily on how population-genetic processes affect phylogenetic inference, but not as much on the nature of the pop-gen processes themselves, or how they arise in natural populations and interact with our conception of species and speciation.

2) The authors don’t take a broad time horizon. Despite talking about the Tree of Life, they focus mostly on the present day, with little discussion of fossils, ancient DNA, or paleobiology. This is consistent with a focus on genomics, but doesn’t represent the whole of the Tree of Life.

3) The authors state problems that face our collective efforts, but don’t really address the underlying structural issues that give rise to those problems, such as institutionalized focus on impact in developing nations and many western institutions. Some parts read simply as a list of problems, and then the statement “this should change!” What concrete steps could we take to, for instance, foster an international dedication to free and open data sharing, while preserving novelty and impact for new datasets?



Specific sentences that might be improved:

"Phylogenomics and macroevolution represent two ends of a research spectrum, with one end focusing on building phylogenies and the other end on using them.”

This statement seems problematic; I don’t think most researchers would find any dichotomy or opposition between phylogenetic inference and phylogenetic comparative methods. They are also very close to each other compared to, say, macro ecology. Could this be phrased differently?

"More generally, evolutionary biologists have not yet thought much about the type of new questions that we are going to be able to address if we are given genomic data at the tips of all species from a phylogeny.”

This also seems unwarranted; I’m certain that very many people have thought very deeply about this. Perhaps its more accurate to say that a large literature does not exist to give a framework for this perspective?

"While such approaches elicit a culture clash between those who laboriously build trees and those that simply use them, there are other approaches stemming from macroevolution that are less offending to phylogeny builders.”

Again, I’m not exactly sure what the authors are trying to say here, but perhaps it could be phrased more delicately? First, I’m not sure that there’s a huge divide between “builders” and “users;” certainly, they are commonly the same group, with very few people assuming only one of those labels. Second, it seems like the authors are implying that Jetz, Tonini, and Kuhn are “users,” and that imputation offends “builders?” I assure you that all three groups are builders; Tonini was my doctoral student, and my lab is known for building trees if nothing else. Imputation is simply a stop-gap measure to make full-scale phylogenetic datasets available for “use” in a way that compartmentalizes phylogenetic error to the taxonomic level.

Perhaps I’ve misunderstood the intent here, but it could be written differently?

Finally, in this section the authors mention extinct species, but there’s very little, if any discussion devoted to fossil-bearing trees, or tip-dating. The authors might discuss Pyron (2011), Ronquist et al. (2012), or many of the recent Slater, Matzke, and Wright papers on including fossil tips in molecular phylogenies to estimate the “true” Tree of Life through time, and not simply the present-day living slice. I realize this is a paper on genomics, but researchers will still want fossil-bearing trees even with genomic data. The authors mention DNA from old museum specimens; this begs discussion of ancient DNA and potential fossil-genomics.



Finally, I’m a bit confused by the Dickens quote to open, which refers to the French Revolution and Robespierre’s Reign of Terror. Is this really the worst of times for phylogenomics? Or do we simply have a surfeit of data that we haven’t quite figured out how to exploit fully? Do the authors plan on throwing off the shackles of an entrenched aristocracy and leading royals to the guillotine? If not, perhaps another quote is better?

Perhaps Shakespeare’s _The Tempest_: For phylogenomics, “hell is empty, and all the devils are here!”

Or Mark Twain’s _The Horse’s Tale_: “Herodotus says 'Very few things happen at the right time, and the rest do not happen at all. The conscientious historian will correct these defects.’"

·

Basic reporting

This perspectives piece on “Embracing heterogeneity: Building the tree of life and the future of phylogenomics” provides well rounded summary on current approaches and limitations in phylogenomics and proposes areas of research that need development over the next few years. It will appeal broadly to those working in the field of phylogenomics.
Overall the perspectives piece is well balanced, written and referenced. It gives a good overview of the field and thus I have no major criticisms to offer.

Instead I have some minor comments which will help with the overall readability as well as minor comments on specific examples/topics which could be expanded or included.

Experimental design

no comment as this is a perspectives piece.

Validity of the findings

no comment as this is a perspectives piece.

Additional comments

• The introduction [57-102] is verbose and needs further editing so that it is more concise and better summarises the topics will be discussed for the remainder of the perspectives piece.
• The survey methodology paragraph is not necessary as the details are covered in the acknowledgments.
• There are several acronyms used throughout this perspectives (ILS, MSC, GDL, ARG, and HTS) which makes the content difficult to quickly digest for the reader. A standalone text box that summarises the main acronyms used in the paper would be a helpful addition.

• III data generation and data types in phylogenomics [114-170]. This section does not follow the title heading. A discussion on collection of data types and sequencing approaches was expected but instead the paragraph vaguely discusses taxon versus marker size. Editing this section so that it more adequately reflects what is about to be discussed in the sub headings is advised.
• The statement referring to “large-scale phylogenetic trees are cobbled together from disparate existing sources, even taxonomy, but often without hard data being the placement of many species” needs further clarity [122-125]. It is generally acquisitive without clear examples from the literature.
• A comment on technical errors that accumulate in sequences due to different sequencing chemistries is advised as well as commenting on varied sequencing depths across species which impacts the ability to reconstruct the genome and reliably call nucleotides.

• Data generation and marker development [171-278].
Avoid using the word “impossible” in the following statement “the non-coding genome is difficult if not impossible to align and analyse” [line 183-184]. The MultiZ vertebrate alignments available through Ensembl and UCSC demonstrate that alignments of reasonable proportions of the non-coding regions are possible. Rather than focusing on what cannot be done now, the authors could discuss advances that are underway in understanding the genetic code of the non-coding genome and mention that it will take development of new methods and approaches to interpret the evolutionary history of these genomic regions. For example, while there is low levels of sequence conservation, regulatory conservation and conserved 3D structures show higher levels of conservation and could be used for comparative genomics purposes in the future (see Berthelot C et al 2018,doi:10.1038/s41559-017-0377-2, Villard D et al 2015 (doi: 10.1016/j.cell.2015.01.006) and Pope et al 2014 doi:10.1038/nature13986).
• An expanded comment on the limitations of transcriptomic data is needed [185]. One of the major shortcomings is that the transcriptome is subject to tissue specific splicing and tissue specific expression making it difficult to compare across species.

• More taxa versus more loci [238 - 276]. In this section while the authors mention that there will be a jump in the size of phylogenetic datasets, they do not give specific examples of some of the ongoing efforts to increase WGS of taxa e.g. Genome 10K or the BGIs plans to sequence 1 million plant, animal and micro-ecosystem genomes.


• Filtering heterogeneous phylogenomic data sets [277]. Mention that balanced taxon sampling will help overcome the effect of outliers.
• Furthermore, there are systematic methods available to detect spurious species (see work by Siu-Ting et al (2015) https://doi.org/10.1093/sysbio/syu066)

• Heterogeneity and Intra individual state polymorphisms [295-378]. The authors discuss markers of phylogenetics which could include rare genomic changes, gene order and synteny, indels and transposition and copy number variations. It would be beneficial here to mention the benefits in congruence across multiple markers. (Rota-Stabelli et al (2011) discusses this ( doi: 10.1098/rspb.2010.0590))

Finally, it would be beneficial to comment on recent specific examples in HTS. PacBio provides long reads which will significantly improve our ability to identify large structural changes in genomes. Advances in sequencing technology such as the HiSeq X or NovaSeq allow for a significant reduction in cost of sequencing which can result in more taxa at a higher coverage. Finally, the Naopore is as hand held device which can be used in the field. These technical advances will have a significant impact on the increase in genomic loci and taxa available for reconstruction of the tree of life.

Reviewer 3 ·

Basic reporting

The review is an extensive and very broad overview of genomic data and the development of Multiple Species Coalescent models, with a particular focus in its uses in biodiversity research (species delimitation, phylogeography and species sampling) in phylogenomics. I think it is timely and it will be of interest mainly to biologists and bioinformaticians that have certain base knowledge of the methods (I wouldn’t say it is intended for a novice audience). It is within the scope of the journal (as a Literature review).

There has been a few recent reviews on this topic, but more into the actual phylogenomics methods (supertrees, supermatrix and models of nucleotide substitution in phylogenetics). The present review focuses primarily on multispecies coalescent models, and in this particular topic there is another recent review (more like a opinion piece) made by one of the authors (S. Edwards) published in Molecular Phylogenetics and Evolution in 2016 (https://doi.org/10.1016/j.ympev.2015.10.027). But the focus of both differ substantially, with the 2016 paper addressing a criticism made by other authors on transcriptomic used in MSC and current work being an actual literature review.

The authors address who their target audience is and their motivation for the paper. The only criticism I could raise on this point is that it is not entirely clear who they mean in the phylogenomics community, as the paper mainly concerns the Multispecies Coalescent model. For a phylogenomics review, I feel that this is lacking of a very important component: the traditional methods and the vast number of models that have been made in this topic of research, as well as other latest developments in different aspects that are of importance to current phylogenomic analyses (alignments and supermatrices methods of phylogenetic reconstruction and models used in these). The "Concepts and models in phylogenomics" and “Bypassing full likelihood models by relying on summaries of the coalescent process” sections are heavily focused on Multispecies Coalescent Models, and dismiss the traditional methods and models still very widely used and comprise the vast amount that lead to the field of phylogenomics in the last 15 years. For instance, recent advances in supermatrix models and methods include the use of codon models of evolution in Maximum likelihood (see for example IQ-tree), also there’s the CAT model implemented in Phylobayes (Lartillot and Philippe 2004) with its MPI version; also when talking about heterogenous models of evolution there’s P4 (Foster 2004), just to mention a few. It seems to me they try to “get off the hook” from covering the other methods with the statement: "Our goal is not to provide a complete overview of phylogenomic and phylogeographic research, but rather present a number of conceptual and practical aspects that we feel are essential to keep the momentum that these fields have gained during recent times." At the moment this statement would mislead an audience that works traditionally with supermatrix methods in phylogenomics, as MSC is not all of phylogenomics. I think it should be stated from the very beginning (including the title and abstract) that the focus of the paper is about the future of multispecies coalescent methods in phylogenomics.

Another point: the use of the term “heterogeneity”. I think this could also lead to some confusion to anyone who works in phylogenomics because there are also heterogenous models of evolution, compositional heterogeneity, site heterogeneity (by the way, neither mentioned in this review). I know, after reading this review, this is not what was meant, but perhaps it would be worth making a small annotation somewhere in the introduction that they do not mean any of these other topics.

Another issue I would raise is that most of the focus of the paper is on evolution at the tips of the tree (or relatively recent divergences), with very little mention on how this could affect deep level evolutionary relationships. In fact, there’s several recent advances (some cited in the first part of this review) addressing deep nodes and their divergence times that even with large-scale data remain controversial. All of these use genomic scale data.

The section on "Tree-free methods of character evolution" - I think it moves the paper out of the scope of phylogenomics here. It's nicely done, don't get me wrong but there’s no relation of this to genomic data and, I just think that it is a bit out of place in an already very broad paper.

In terms of structure and citations, sources are adequately cited and paraphrased as appropriate as far as I could see. The review is mostly well organized into coherent paragraphs and subsections. I would point out here that all the sections starting from “Heterozygosity and Intra-Individual Site Polymorphisms” to “Copy Number Variations” should be listed under a new different subsection with its new name (not Data generation and marker development) where it currently is.

Experimental design

I have no comments.

Validity of the findings

The conclusion identifies unresolved questions and future directions, namely on curation of metadata associated to specimens and sequencing data, potential new ways to integrate other types of data in the future. Having made such a thorough review on the types of genomic data in the beginning, it lacks any mention of it in the end.

In my opinion, the review mainly focuses on 3 topics: Types of genomic data and their caveats (mostly on the MSC), Multiple Species Coalescent models and some potential uses in biogeography and expanding the species sampling in the tree of life. The authors have done a very nice job to collect an exhaustive amount of information and point out numerous aspects in these topics, some very interesting remarks there too, but it feels like they tried to put too many different things together and these are so numerous, that they become inconsequential and lost in the paper (for example the literature review to show the increase in number of loci vs number of species and the other methods that are not MSC). There’s barely any mention of this in the end and it makes me wonder why you would go through such trouble when it is barely mentioned again. In addition to that, the amount of topics is so broad that it becomes a bit difficult for the reader to pin down what are the main messages here. There is an attempt to bring it together, but I missed the point of discussing the different caveats of genomic data along with the methods, and then along with the tree of life. I think there is potential for improvement in connecting those main ideas and rounding this paper.

Additional comments

Overall I think it is a great effort done to try to put together many topics that can impact MSC in phylogenomics, and the future of how this could extend to biodiversity. Many interesting points raised. I am of the opinion it should be published and that it will make a nice contribution to the field of phylogenomics.

I have raised some points above, that are made with the hope that it will improve the manuscript and that it won't be misleading (in reference to dismissing other methods apart from MSC). I have also listed some minor corrections that need to be addressed in an attached pdf.

Annotated reviews are not available for download in order to protect the identity of reviewers who chose to remain anonymous.

---

## Round 0.2 · Minor Revisions

Thank you for your re-submission. Could you please have a look at the comments made by the reviewers and in particular address issues around typos, or missing information.

Some of the issues that have been raised by the reviewers I think could be best addressed by including a short section (or box) towards the end of the review which outline the outstanding questions/challenges in the field. This will allow a useful summary of the reader of what future work will be needed and help address issues raised by the reviewers, for example as said by reviewer 1 "that this section is a list of “good ideas” without any good ideas; there are few, if any, really strong suggestions for HOW any of these problems should be addressed".

I don't think that your review needs to give answers to outstanding questions, but it should highlight them clearly, indicating where future research should go and to allow your extensive review to become a reference for the field.

If you could address this, I would be happy to accept the manuscript without further review.

·

Basic reporting

no comment

Experimental design

no comment

Validity of the findings

no comment

Additional comments

I’ve reviewed this again and consulted the other reviews and response, and I still have some lingering reservations about the structure of the review. It would be easy to nitpick a review of this length forever, and I’m not trying to do that. But I still want to re-iterate my concerns, and hope the authors can address some of them.

In no particular order:

First sentence; is it really an embarrassment of riches? Isn’t one of the points made that we lack the full datasets (i.e., whole genomes) that would allow for easy a posteriori marker selection? Do we really need a “snappy” quote in the introduction at all?

The authors stated that they didn’t want to review population genetic processes, because they are focusing on phylogenomics. However, as noted by another reviewer, it seems like they want to have it both ways. The authors talk about how these processes might mislead phylogenomic analyses, AND they talk about species delimitation and concepts. Yet, they still don’t discuss HOW the population-genetic processes interact to bridge these scales. There’s a lot of talk about the population-phylogeography-phylogeny continuum, but it comes off as a bit shallow since the actual mechanistic effects aren’t introduced to link HOW population processes actually affect genealogical signal.

Related to the above, the citation patterns are a bit odd; not citing Avise et al. 1987 or Edwards and Beerli 2000 (among other classics) in the intro section, but waiting until much later.

Also related to (2), the authors focus on the MSC because it forms the basis of many current methods, but that’s a small slice of the relevant pop-gen literature. I also feel like there’s not a detailed enough description, again, of HOW the various pop-gen processes mentioned actually affect the outcome of these methods. Part of that is due to it not being known. But it leaves the reader wanting, I think, for a fuller understanding.

The issue regarding pop gen and the MSC is part of a larger issue here, I think, which is that the authors are trying to summarize too broad a range of material, without sufficient depth in parts of it. I realize this piece originated from a wide-ranging workshop, but the really valuable depth is concentrated in a few sections, while the other leave much to be desired. My main suggestion here is to consider trimming several of them. I’ll highlight the ones that seem, to me, to be most superfluous:

3.3.2-3.3.3: This material is rapidly becoming irrelevant as we move closer to whole genomes. It also illustrates the false dichotomy of taxa vs loci, as the number of loci is fixed (i.e., the genome), and is being rapidly approached.

3.3.4: it seems like there’s very little detail or depth here; the latter part is simply a list of methods and citations with no discussion of how they function.

3.5 This short section again just simply lists several sequencing technologies…

Section 4-4.3: THIS is the really valuable meat of the review. There’s a lot of detailed information here, reflecting the expertise of the authors. I found this whole section to be immensely valuable; just reading through it during review has already changed my perception of the direction of the field, illuminated some patterns in my own datasets, and altered the course of my research program.

Section 5: This material seems disconnected from phylogenomics, and is primarily just reiterating information found in numerous other reviews. The only exception is 5.1, talking about genomes and phenotypes. But do we really need another review discussing OU and BM models?

Section 6: This is a high-variance section for importance and quality. For one, the authors are still criticizing data imputation for building large trees, but then champion the Open Tree of Life, which is imputed!

6.3 what does mapping have to do with phylogenomics?

6.4.1 This section has a lot of sentences that seem obvious and redundant, without offering much that is new for this review. For instance: "In general, we advocate for following best practices of data management and publication to ensure the quality and utility of phylogenomic data and their associated biological information (see Costello & Wieczorek, 2014 for a review).”

Do we really need a sentence where the authors say that they feel practitioners in the field should follow best practices? Was that in question?

Similarly, and I don’t want to seem petty, but: "Recent genomic techniques now allow successful results in obtaining valuable DNA data from museum specimens (e.g., Staats et al., 2013; Hykin, Bi & McGuire, 2015; McCormack, Tsai &mFaircloth, 2016; McCormack et al., 2017; Ruane & Austin, 2017), and here, we advocate for routine use of these resources to enhance research in phylogenomics and phylogeography and increase awareness and usefulness of natural history collections.”

This sentence just says that biodiversity scientists should use available natural history collections and molecular methods to conduct phylogenomic research… Does that really need to be said?

I feel like my original concern still stands; that this section is a list of “good ideas” without any good ideas; there are few, if any, really strong suggestions for HOW any of these problems should be addressed.

It’s also unclear how this relates to the MSC and phylogenomics; it’s almost as if there are two or three distinct, almost unconnected reviews being conducted simultaneously. The section 6 material is certainly not reflected in the abstract or the introduction.

My suggestion would be to trim a great deal of material, focusing on developing Sections 1, 2, and 4. I don’t want to stand in the way of this publication; I’m not trying to stall it out. Hopefully, editorial guidance will provide a clear path forward.

Reviewer 3 ·

Basic reporting

The article has improved from the previous version I read. I think it reads more clearly, and the new sections that they have organised it into helps to group the topics better. I like the box with definitions added.

Experimental design

No comment.

Validity of the findings

My opinion is that the article just needs a few minor things to modify/ address before being accepted for publication.

1) First sentence: "Phylogenomics, the practice and molecular methods focused on building the Tree of Life (ToL), is currently suffering from an embarrassment of riches. " -- Well, I guess this is the authors' choice, but I wouldn't see having so many options in phylogenetics methods and data a hindrance, on the contrary, that's how scientific rigour should be!

2) P11: "This a posteriori selection would be carried out after assessing desired phylogenetic and biological properties of a wide array of markers for which the data are already in hand." -- Just to make it all homogenous, and as done with the "a priori" and in the response to the review, you should add the small definition here with parenthesis after a posteriori, i.e: "a posteriori (after data generation)".

3) P12 end - P13 top: "Some researchers favor “horizontal” data matrices, wherein the number of loci far exceeds the number of taxa, whereas other researchers favor “vertical” matrices, where many taxa are analyzed at just a few (1-5) loci. Whereas the PCR era of phylogenetics was often dominated by vertical matrices, HTS is allowing data matrices to become more horizontal (Fig. 2)" -- It would be good if the authors added just a statement about the problems that this brings with missing data issues and poor overlap all contributing to either misleading or poor phylogenetic resolution, given the numerous discussions revolving this issue across all phylogenetics.

4) P23 - "loci consistent with the MSC model can be identified (e.g., separated from loci with horizontal gene transfer) using the CLASSIPHY (Huang et al., 2017). " -- Looks like the word "the" before "CLASSIFY" is redundant, or did you miss something here?

5) P37 - "Another point of possible improvement is in dealing with new sequences to be added to a previously large dataset: should the analysis start from scratch, or could there be substantial time gains by letting those sequences find their placement in the phylogeny ‘on the fly’ (e.g., Siu-Ting et al., 2014)?" -- Sentence should end in "."

6) Figure 3 legend. "This implies that tree building without considering the multispecies coalescent could, in this case, lead to erroneous estimation of tree topology and divergence times." -- There is an issue with the last sentence here, it doesn't follow logically from what is explained in the rest of this caption, which is all about violations to the MSC. I would suggest removing it.

7) typo of of the word "homoeologous" in several instances.

8) Authors should check that all citations in the text are included under the Literature Reference list.

9) In my last review, I raised the following issue:

"Another issue I would raise is that most of the focus of the paper is on evolution at the tips of the tree (or relatively recent divergences), with very little mention on how this could affect deep level evolutionary relationships. In fact, there’s several recent advances (some cited in the first part of this review) addressing deep nodes and their divergence times that even with large-scale data remain controversial. All of these use genomic scale data."

Response received from authors: "Following a similar comment by reviewer # 1, we have now added a section extinct taxa, fossils, and tip dating to cover that dimension that is beyond the reach of genomic data. We focus this paragraph on how the integration of both sources of data can be used toward the Tree of Life."

-- I would like to clarify that I did not mean any of the issues that you mention in the response and that you have now added as a new section (namely including fossils and unsampled taxa). I actually meant that what was lacking was a mention of how useful could including genome data, or including more genes or more taxa used in a MSC context actually help when addressing deep nodes in the ToL. Is it applicable to use at that level of divergence (deep nodes)? Numerous studies which you have cited in your introduction and section 3.3 (and works that these papers cite) feature these unresolved/disputed areas in the ToL (for example the sister group to animals, the sister group to arthropods, the position of turtles in the reptile+bird group, just to name a few) in spite of the increasing amount of data that these works include (i.e. either more taxa sampled or more genes included). I think that this is missing somewhere here in the review, and it could be a pertinent point to add in the 3.3 section, perhaps in the "3.3.3 More taxa versus more loci" or the "3.3.4 Filtering heterogeneous phylogenomic data sets" sections of this paper.

Additional comments

The article has improved from the previous version I read. They have addressed the majority of my questions and comments and what perhaps was my biggest concern, which was to clarify the audience and the type of methods they intended to cover (mainly phylogenomics and mostly in the realms of MSC). I think it will make a great contribution to the current literature in phylogenomics as it addresses pertinent and timely problems. It's nearly there, just a few minor things that in my opinion will improve the manuscript.

---

## Round 0.3 · accepted · Accept

Thank you for your updated manuscript. I am happy to recommend its acceptance in its present form.

#